# The mediodorsal pulvinar coordinates the macaque fronto-parietal network during rhythmic spatial attention

Ian C. Fiebelkorn[1], Mark A. Pinsk[1] & Sabine Kastner[1,2]

Spatial attention is discontinuous, sampling behaviorally relevant locations in theta-rhythmic cycles (3–6 Hz). Underlying this rhythmic sampling are intrinsic theta oscillations in frontal and parietal cortices that provide a clocking mechanism for two alternating attentional states that are associated with either engagement at the presently attended location (and enhanced perceptual sensitivity) or disengagement (and diminished perceptual sensitivity). It has remained unclear, however, how these theta-dependent states are coordinated across the large-scale network that directs spatial attention. The pulvinar is a candidate for such coordination, having been previously shown to regulate cortical activity. Here, we examined pulvino-cortical interactions during theta-rhythmic sampling by simultaneously recording from macaque frontal eye fields (FEF), lateral intraparietal area (LIP), and pulvinar. Neural activity propagated from pulvinar to cortex during periods of engagement, and from cortex to pulvinar during periods of disengagement. A rhythmic reweighting of pulvino-cortical inter-actions thus defines functional dissociations in the attention network.

[1] Princeton Neuroscience Institute, Princeton University, Princeton, NJ 08544, USA. [2] Department of Psychology, Princeton University, Princeton, NJ 08544, USA. Correspondence and requests for materials should be addressed to I.C.F. (email: iancf@princeton.edu)

Spatial attention leads to enhanced sensory processing and improved behavioral outcomes (e.g., higher hit rates and faster reaction times)[1,2]. Classic studies of spatial attention assumed that its neural and behavioral effects were continuous during attentional deployment. Recent studies, however, have instead demonstrated that spatial attention is discontinuous, sampling the visual environment in theta-rhythmic cycles (3–8 Hz)[3–8]. We recently linked rhythmic sampling to theta oscillations in frontal and parietal cortices[4,5]. These theta rhythms organize neural activity into alternating attentional states associated with either enhanced or diminished perceptual sensitivity[4]. We have proposed that theta-rhythmic sampling thus reflects alternating periods of either (i) engagement at the presently attended location or (ii) relative disengagement, with periods of disengagement likely associated with an increased probability of attentional shifts[4]. In this way, theta-rhythmic sampling provides spatial attention with critical flexibility, offering windows of opportunity when it is easier to disengage from the presently attended location and move to another location. But how are these functionally defined attentional states coordinated across the large-scale network that directs spatial attention?

Research into the neural basis of spatial attention has largely focused on cortical contributions, particularly from frontal and parietal cortices[9,10]. Microstimulation and inactivation studies have shown that these higher-order cortical regions generate modulatory signals that are fed back to sensory cortex[11–13], boosting sensory processing at behaviorally relevant locations[2]. Recent research has challenged this cortico-centric view[14]. Several studies have shown that a subcortical structure, the pulvinar nucleus of the thalamus, regulates cortical activity during spatial attention[15,16]. For example, pulvino-cortical interactions seem to facilitate communication between visual cortices (i.e., V4 and TEO) by aligning the phase of alpha/low-beta activity (8–15 Hz).

The pulvinar is the largest nucleus in the primate thalamus, yet its functional significance has remained largely unknown[17,18]. Lesion and inactivation studies have indicated that the pulvinar plays a critical role in mediating spatial attention. For example, lesions of the pulvinar can result in symptoms that are similar to hemineglect following lesions of parietal cortex[19,20]. Behavioral impairments in human patients are particularly strong when distractor stimuli compete for attentional resources[21,22]. Wilke et al.[23], who used muscimol to inactivate the pulvinar, reported similar behavioral impairments in monkeys. That is, the animals demonstrated diminished exploration of the contralesional visual hemifield during inactivations, particularly when stimuli appeared in both hemifields.

The ventral pulvinar is ideally positioned to regulate interactions across the visual system based on its connectivity with cortex[15,17]. Whereas ventral regions of the pulvinar are largely interconnected with the visual cortex[24–27], dorsal regions are largely interconnected with higher-order cortical regions, including the frontal eye fields (FEF) and the lateral intraparietal area (LIP)[28–30]. The dorsal pulvinar is therefore an ideal candidate for influencing cortical interactions in the attention network. Few studies, however, have measured neuronal responses in the dorsal pulvinar during spatial attention[31,32], and its functional role in the attention network has not yet been defined.

Here, we investigated whether pulvino-cortical interactions coordinate the theta-rhythmic attentional states that seemingly shape spatial attention. We therefore simultaneously recorded single-unit activity and local field potentials (LFPs) from FEF[33], LIP[34], and the pulvinar[17], while monkeys performed a spatial attention task that has previously been shown to promote theta-rhythmic sampling[3]. We specifically targeted the mediodorsal pulvinar (mdPul), which receives overlapping projections from frontal and parietal cortices (Fig. 1)[28]. The present results show

that the source of functional connectivity between mdPul and higher-order cortices—which occurs in the alpha/low-beta band (10–20 Hz)—shifts with attentional state, with (i) mdPul being its source during the theta-dependent state associated with engagement (and enhanced perceptual sensitivity) (ii) and LIP being its source during the theta-dependent state associated with relative disengagement (and diminished perceptual sensitivity). We propose that theta-dependent changes in alpha/low-beta activity reflect a functional re-weighting across hubs of the attention network. This rhythmic re-weighting alternately favors brain regions and pathways that promote either attentional sampling or shifting.

## Results

**Procedure.** Two male monkeys (*Macaca fascicularis*, 6–9 years old) performed a covert spatial-cueing paradigm based on the Egly-Driver task[4,35]. A peripheral spatial cue indicated where a subsequent, low-contrast visual target was most likely to occur (78% cue validity; Fig. 2). The animals maintained fixation, holding down a lever to begin each trial and releasing the lever to indicate visual-target detection. We specifically focused on neural activity during the time period after the cue-evoked response (i.e., the visual-sensory response) and until target onset, which generally satisfied methodological assumptions of stationarity (see Methods). During this cue-target delay, the animals were deploying spatial attention at the cued location. For between-condition comparisons, we measured neural activity when receptive/response fields overlapped the cued location relative to when receptive/response fields overlapped the non-cued location positioned on a second object[4,35] (see ref. 4 for previously reported behavioral evidence of attentional deployment at the cued location).

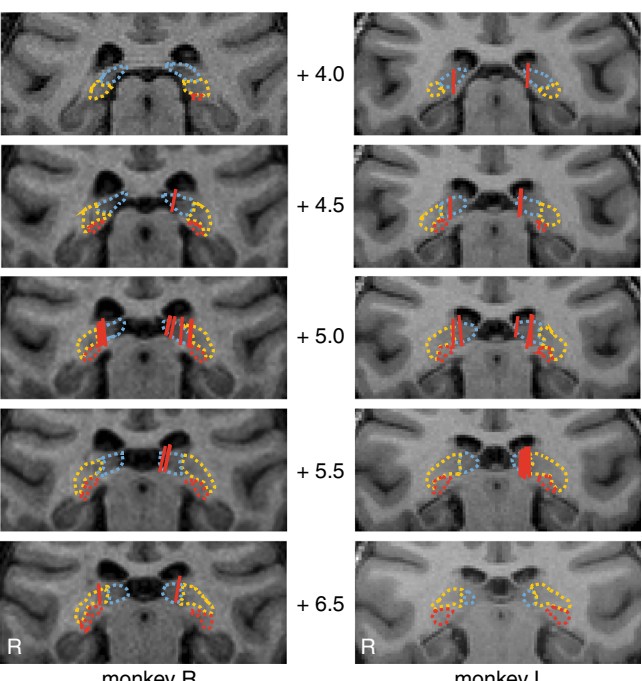

**Fig. 1** Penetrations targeting the atlas-defined mediodorsal pulvinar (mdPul). We previously published representative penetrations for FEF and LIP[4]. Here, a red line depicts each penetration. The numbers positioned between images from each of the animals represent anterior–posterior distances from the interaural line (mm). Atlas-defined pulvinar subdivisions: dashed blue = medial pulvinar; dashed yellow = lateral pulvinar; dashed red = inferior pulvinar. The white R = right hemisphere

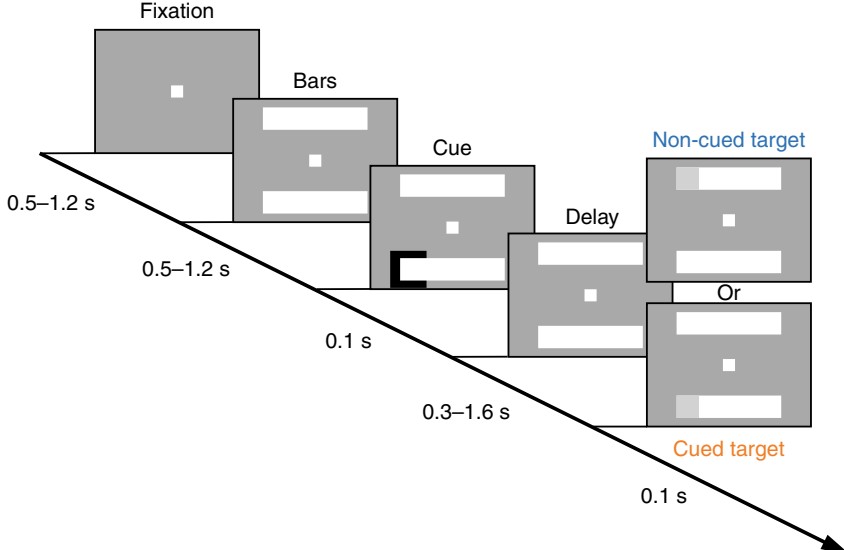

**Fig. 2** A schematic of the behavioral task. The animals pressed a lever to begin each trial and maintained central fixation. A spatial cue indicated, with 78% cue validity, where the visual-target was most likely to occur. Following a variable cue-target delay, a low-contrast visual target was either presented at the cued location or at a non-cued location. The animals responded by releasing the lever. We also included catch trials to track false alarms

**mdPul contributes to the maintenance of spatial attention.** Given that the present study is the first to investigate attention-related function in mdPul, we started by characterizing neuronal responses in mdPul and then comparing them with neuronal responses in FEF and LIP. That is, we first investigated whether mdPul is a functional hub of the attention network. Supplementary Table 1 displays the number of neurons in each region of interest (ROI) that demonstrated task-related activity. Neurons in frontal and parietal cortices are typically classified based on their response profiles as visual (i.e., sensory-related responses), movement (i.e., saccade-related responses), and visual-movement (i.e., sensory and saccade-related responses) types (see Methods and ref. [4]). We recorded from a total of 224 neurons in mdPul, with 52 neurons (23.2%) demonstrating significantly increased spiking activity after presentation of the cue ($N = 20$ visual-sensory neurons), the target ($N = 13$ movement neurons), or both the cue and the target ($N = 19$ visual-movement neurons). A small population of neurons demonstrated significantly decreased task-related activity ($N = 9$). In comparison to mdPul, 98/238 neurons (41.2%) in FEF and 98/259 neurons (37.8%) in LIP had significantly increased task-related activity. Although the overall percentage of task-responsive neurons was higher in both FEF and LIP, the distributions of visual, movement, and visual-movement neurons (among task-responsive neurons) were similar across the three ROIs.

Figure 3a shows normalized population spiking, time-locked to the cue, and averaged across all neurons with a significantly increased visual-sensory response (i.e., visual and visual-movement neurons). Figure 3b shows normalized population spiking, time-locked to the target and averaged across all neurons with a significantly increased movement response (i.e., movement and visual-movement neurons). In both cases, the responses of mdPul neurons were more similar to LIP than to FEF. For example, spiking activity during the delay did not differ between mdPul and LIP (Wilcoxon rank-sum test, $p = 0.386$), but was significantly higher for FEF neurons (FEF vs. mdPul, Wilcoxon rank-sum test, $p = 0.013$; FEF vs. LIP, Wilcoxon rank-sum test, $p = 0.0008$).

Previous studies on cortical hubs of the attention network have shown that only neurons with visual-sensory responses (i.e., visual and visual-movement neurons)—and not movement

neurons—demonstrate significant attention-related spiking during the cue-target delay[36,37]. Here, we demonstrate that mdPul neurons follow the same pattern, with only visual and visual-movement neurons having significant spiking during the cue-target delay (Supplementary Figure 1). We therefore only used neurons with visual-sensory responses (i.e. visual and visual-movement neurons) to determine whether delay spiking in mdPul was significantly higher when receptive fields overlapped the cued location relative to when receptive fields overlapped the non-cued location. For this comparison (i.e., cued vs. non-cued), we averaged spike rates across a 450-ms window, just prior to target presentation, revealing significantly higher delay spiking when receptive fields overlapped the cued location (Wilcoxon rank-sum test, $p = 0.011$). We previously performed the same analysis to demonstrate significantly higher, attention-related delay spiking in FEF and LIP[4]. The present findings therefore demonstrate that mdPul—like FEF and LIP—contributes to the maintenance of spatial attention at the cued location and appears to be an integral part of the attention network.

**mdPul contributes to theta-rhythmic sampling.** Recent evidence has demonstrated that spatial attention samples the visual environment in theta-rhythmic cycles[3–8]. We previously linked this theta-rhythmic sampling to the phase of theta oscillations in FEF and LIP[4]. Theta rhythms organized neural activity in these cortical hubs of the attention network into two alternating attentional states, associated with either (i) better visual-target detection at the cued location (i.e., during the "good" theta phase) or (ii) worse visual-target detection at the cued location (i.e., during the "poor" theta phase). Supplementary Figure 2 shows LFP power in mdPul, from 3 to 60 Hz, demonstrating a clear peak in the theta range (after removing the $1/f$ component). Having previously confirmed mdPul as a functional hub of the attention network (Fig. 3), we next measured whether theta phase in the LFP influenced the likelihood of detecting a low-contrast visual target (see Methods). We found a significant phase-detection relationship (permutation test, $p < 0.0009$), with a peak at 5 Hz (Fig. 4a), demonstrating that mdPul—like FEF and LIP—has a role in theta-rhythmic sampling during spatial attention. That is, theta-band activity in mdPul is also associated with alternating periods of either better or worse visual-target detection.

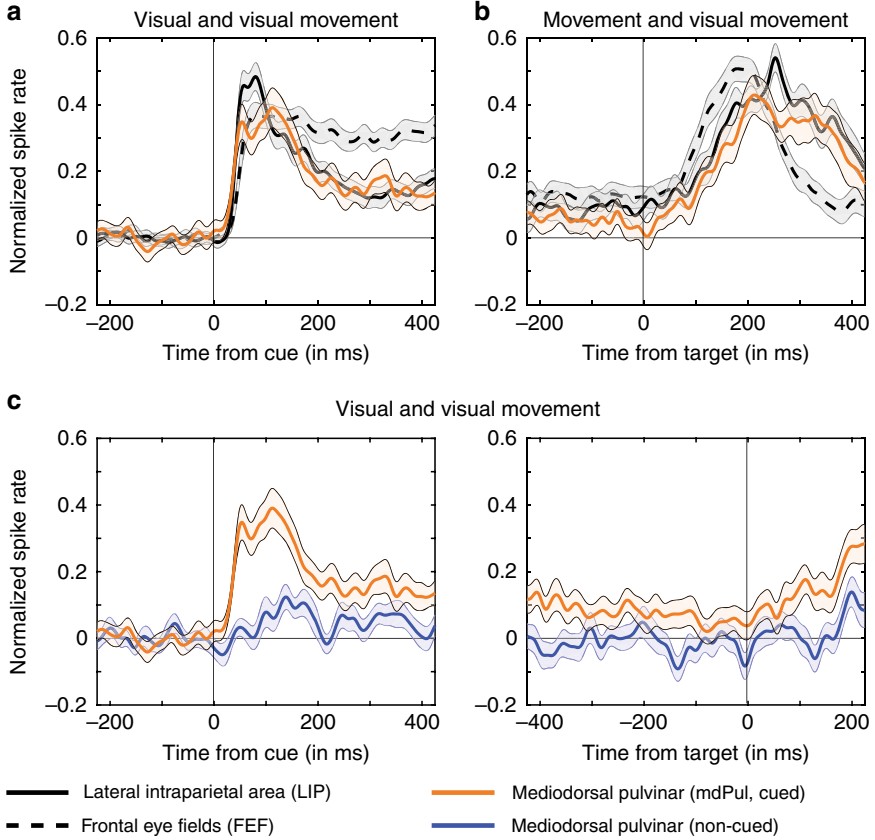

**Fig. 3** mdPul contributes to the maintenance of spatial attention at a cued location. **a** Normalized, cue-locked spike rates, averaged across the population of neurons with significantly increased visual-sensory responses (i.e., visual and visual-movement neurons) when receptive fields overlapped the cued location. **b** Normalized, target-locked spike rates, averaged across the population of neurons with significantly increased movement responses (i.e., movement and visual-movement neurons) when receptive fields overlapped the cued location. **c** Normalized spike rates in mdPul, time-locked to either the cue (left) or the target (right), both when receptive fields overlapped the cued location (orange) and when receptive fields overlapped the non-cued location (blue). These plots are averaged across all mdPul neurons with a significantly increased visual-sensory response (i.e., visual and visual-movement neurons). Shaded regions around the lines represent SEs

In FEF and LIP, we previously observed significant phase-detection relationships not only in the theta band but also at higher frequencies (i.e., in the alpha [9–15 Hz], beta [15–35 Hz], and gamma [>35 Hz] bands), with different frequency bands characterizing the two rhythmically alternating attentional states[4]. For example, the theta-dependent state associated with better visual-target detection was characterized by increases in both (i) FEF-dominated beta activity and (ii) LIP-dominated gamma activity. Beta and gamma activity in FEF and LIP were therefore only predictive of behavioral performance during this "good" theta phase (see ref. [4], Fig. 3e–f). In comparison, the theta-dependent attentional state associated with worse visual-target detection was characterized by an increase in alpha/low-beta activity in LIP. Alpha/low-beta activity in LIP was therefore only predictive of behavioral performance during the "poor" theta phase (see ref. [4], Fig. 3f).

Based on these previous findings in FEF and LIP, we next examined whether higher-frequency activity in mdPul was linked to a specific attentional state. We re-calculated phase-detection relationships in mdPul (from 9 to 60 Hz) after first splitting trials into two bins: (i) a bin centered on the "good" theta phase (at 5 Hz) and (ii) a bin centered on the "poor" theta phase. Figure 4b displays the results, showing a significant link between alpha/low-beta phase (12–16 Hz) and visual-target detection (permutation test, $p < 0.001$). This phase-detection relationship specifically occurred during the "good" theta phase. Notably, this is opposite to the result that we previously reported in LIP (10–18 Hz; see

ref. [4], Fig. 3f), where alpha/low-beta phase was only predictive of visual-target detection during the "poor" theta phase. See Supplementary Figure 3 for corresponding evidence of (i) phase–amplitude coupling (PAC) between theta phase and alpha/low-beta power (11–23 Hz) in mdPul and (ii) previously reported evidence of PAC between theta phase and alpha/low-beta power (9–16 Hz) in LIP[4]. These results demonstrate that increases in alpha/low-beta activity in mdPul and LIP are associated with opposite theta-dependent states, indicating a functional dissociation between these two hubs of the attention network. Alpha/low-beta activity in mdPul is predictive of behavioral performance during periods of engagement at an attended location, while alpha/low-beta activity in LIP is predictive of behavioral performance during periods of disengagement.

**mdPul regulates low-frequency oscillatory activity in cortex.** Previous evidence has shown that the ventral pulvinar regulates alpha/low-beta activity in interconnected regions of visual cortex (i.e., V4 and TEO), possibly facilitating the attention-related exchange of information[15]. In the previous section, we demonstrated that alpha/low-beta activity in mdPul was predictive of behavioral performance (Fig. 4). Here, we asked whether alpha/low-beta activity from mdPul plays a role in regulating neural activity in FEF and LIP, thereby extending previous results[14] from lower-order to higher-order cortex.

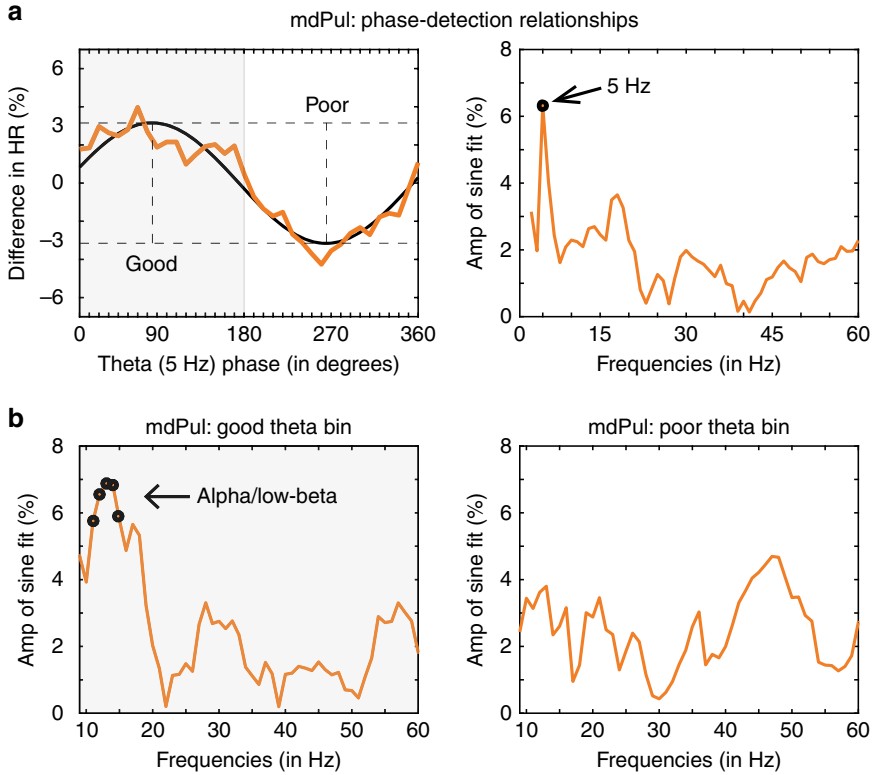

**Fig. 4** The phase of low-frequency oscillatory activity in mdPul is linked to behavioral performance. Hit rates (HRs) were calculated (across recording sessions, $N = 95$) as a function of oscillatory phase when response fields overlapped the cued location. **a** Phase-detection functions were then fit with one-cycle sine waves (left), with the amplitude of that sine wave representing the strength of the relationship between visual-target detection and oscillatory phase. These phase-detection relationships were calculated at different frequencies, from 3 to 60 Hz (right). Phase-detection relationships at higher frequencies (in FEF and LIP) were previously shown to be dependent on the phase of theta-band rhythms[4]. **b** Phase-detection relationships in mdPul were therefore re-calculated after first binning trials into two theta-phase bins: (i) one centered on the theta phase associated with better visual-target detection (i.e., the "good" bin) and (ii) one centered on the theta phase associated with worse visual-target detection (i.e., the "poor" bin). The black dots represent statistically significant results after corrections for multiple comparisons

We first examined spike–LFP phase coupling, which measures whether there is a consistent clustering of spikes around oscillatory phases in LFPs. Figure 5a, b demonstrates significant coupling between spikes in mdPul and the phase of alpha/low-beta activity in both FEF (8–19 Hz) and LIP (15–20 Hz), specifically occurring when both receptive fields and response fields (i.e., the LFP equivalent of the receptive field) overlapped the cued location (permutation test, $p < 0.0009$). When measuring spike–LFP phase coupling, spikes are typically viewed as reflecting regional outputs, while LFPs are viewed as reflecting a summation of regional inputs[38]. The present results are therefore consistent with mdPul driving alpha/low-beta activity in cortical hubs of the attention network.

For mdPul-FEF synchronization, spike–LFP phase coupling was uni-directional, meaning spikes in FEF were not linked to oscillatory phase in mdPul (Fig. 5a). For mdPul-LIP synchronization, however, spike–LFP phase coupling was bi-directional, with attention-related coupling also occurring between spikes in LIP and the phase of low-frequency oscillatory activity in mdPul (Fig. 5b; permutation test, $p < 0.0009$). We later provide evidence that these bi-directional effects (i.e., LIP to mdPul and mdPul to LIP) are temporally dissociable (see Figs. 6 and 7).

We next examined whether alpha/low-beta activity (i.e., the frequency band most prominently linked to pulvino-cortical interactions) mediates cortico-cortical interactions in the attention network. Figure 5c shows spike–LFP phase coupling between FEF and LIP, combining all cue-responsive neurons (i.e., visual and visual-movement neurons). These results revealed no

evidence of coupling between spikes in FEF and alpha/low-beta activity in LIP. In comparison, there was significant coupling between spikes in LIP and low-beta activity in FEF (permutation test, $p < 0.0009$), but that coupling occurred at a higher frequency than coupling between mdPul and either of the two cortical regions (Fig. 5c).

To further investigate whether alpha/low-beta activity shapes interactions between FEF and LIP, we also used a second, broader measure of spiking activity at the population level. That is, we measured between-region PAC, specifically investigating links between alpha/low-beta phase and high-frequency band activity (HFB; 80–200 Hz). HFB is an established proxy for population spiking[39]. Unlike our analysis of spike–LFP phase coupling (Fig. 5)—which examined the responses of single neurons identified as having task-related increases in spiking activity—the present analysis (i.e., with HFB) examined summed responses from the entire population of neurons near the recording electrode. Supplementary Figure 4 displays the results, suggesting that cortico-cortical interactions between FEF and LIP indeed occur in windows defined by alpha/low-beta activity (see also ref. [15]).

To evaluate the directionality of functional connectivity in the attention network, we then calculated Granger causality, estimating the influences of alpha/low-beta activity in mdPul on FEF and LIP (relative to the influences of FEF and LIP on mdPul). Granger causal influences were generally stronger from mdPul to both cortical regions than vice versa (Fig. 6a; permutation test, $p < 0.001$). There was also an alpha/low-beta

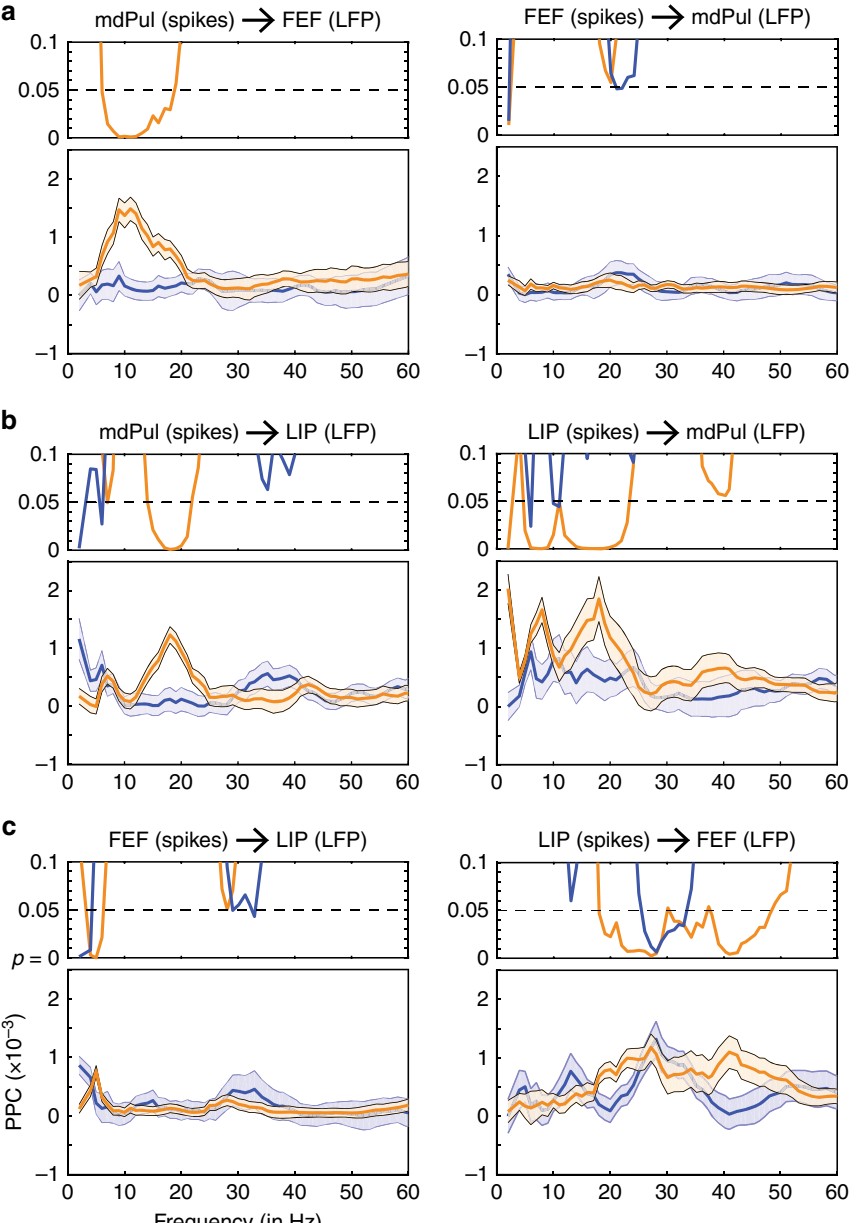

**Fig. 5** Spike–LFP phase coupling demonstrates increased functional connectivity between mdPul and cortex during spatial attention. The top panel of each plot shows *p*-values, indicating whether spike–LFP phase coupling is statistically significant for each condition, cued (orange), or non-cued (blue). The bottom panel compares spike–LFP phase coupling between the two conditions (cued vs. non-cued). Panel **a** examines spike–LFP phase coupling between mdPul and FEF, panel **b** examines spike–LFP phase coupling between mdPul and LIP, and c examines spike–LFP phase coupling between FEF and LIP. These plots are based on data from all task-responsive neurons (see Supplementary Table 1). Shaded regions around the lines represent SEs

peak (at 14 Hz) in Granger causal influence from LIP to FEF. The magnitude of that peak, however, suggests that LIP had a weaker influence on FEF than did mdPul. Supplementary Figure 5 displays conditional Granger causality, which was based on a subset of recording sessions when response fields were aligned across all three ROIs. Conditional Granger causality estimates the influence of one region (X) on another (Y), while accounting for the influence of a third region (Z). The general pattern of results did not change when considering all three ROIs simultaneously rather than in pairs (Fig. 6a). Granger causal influence therefore corroborates our previous interpretation of spike–LFP phase coupling. That is, mdPul regulates alpha/low-beta activity in cortical hubs of the attention network (Figs. 5 and 6; Supplementary Figure 5).

**Pulvino-cortical interactions define functional dissociations**. We have thus far shown that theta-band activity in mdPul—like in FEF and LIP[4]—is associated with alternating periods of either enhanced or diminished perceptual sensitivity (i.e., theta-rhythmic sampling), with increased alpha/low-beta activity in mdPul specifically linked to periods of enhanced perceptual sensitivity (Fig. 4; Supplementary Figure 3). We have also shown that mdPul regulates alpha/low-beta activity in cortical hubs of the attention network (Figs. 5 and 6). We next examined whether pulvino-cortical interactions differed between (i) the theta-dependent attentional state associated with better visual-target detection (i.e., during the "good" theta phase) or (ii) the theta-dependent state associated with relatively worse visual-target detection (i.e., during the "poor" theta phase). We therefore re-calculated each of our

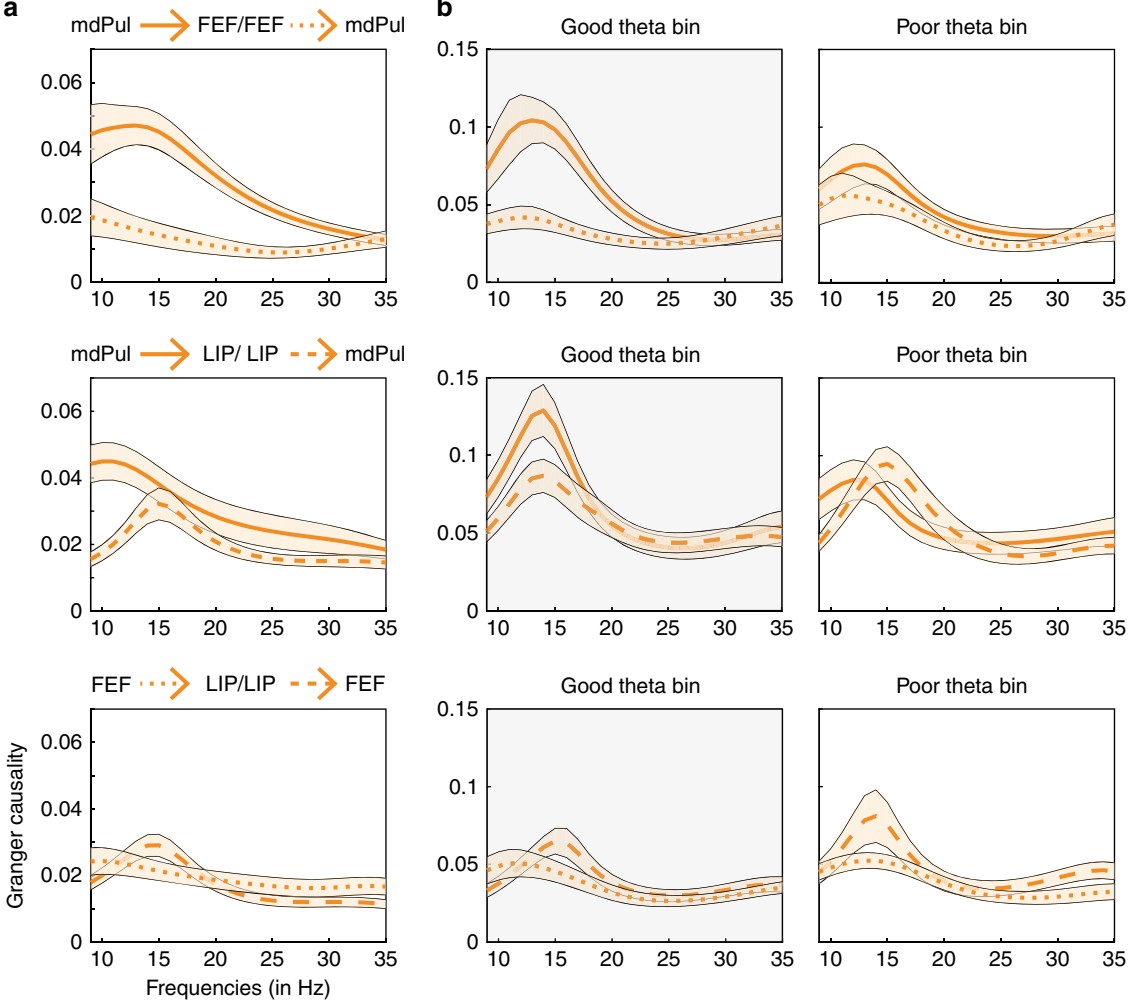

**Fig. 6** Granger causal influence indicates that mdPul regulates alpha/low-beta activity in cortex. **a** For each pair of ROIs, Granger causal influence (model order = 8) was based on all recording sessions with overlapping response fields (mdPul vs. FEF, N = 51; mdPul vs. LIP, N = 57; FEF vs. LIP, N = 67). Supplementary Figure 4 instead shows conditional Granger causality for a subset of sessions when all three ROIs had overlapping response fields (N = 31). Panel **b** shows Granger causal influence after binning based on theta phase ("good" vs. "poor"). mdPul specifically regulates cortical activity during periods of relatively better visual-target detection (i.e., during the "good" theta phase). Shaded regions around the lines represent SEs

between-region measures (i.e., spike–LFP phase coupling and Granger causality) after first binning trials based on theta phase, with one bin centered on the "good" theta phase and the other centered on the "poor" theta phase. The results consistently point to a functional dissociation between mdPul and LIP.

Figure 7a shows spike–LFP phase coupling between mdPul and LIP as a function of theta phase (at 5 Hz). Coupling between spikes and alpha/low-beta activity was generally dependent on the theta phase (permutation test, $p < 0.001$), regardless of directionality (i.e., spikes in mdPul coupled to phase in LIP and vice versa). However, the present results demonstrate that spikes in mdPul were specifically coupled to alpha/low-beta activity (12–18 Hz) in LIP during the "good" theta phase, consistent with mdPul driving alpha/low-beta activity during periods of relatively better visual-target detection. Supplementary Figure 6 provides evidence that spikes in mdPul were similarly coupled to alpha/low-beta activity in FEF during the "good" theta phase. In comparison, spikes in LIP were coupled to alpha/low-beta activity (14–24 Hz) in mdPul during the "poor" theta phase (Fig. 7a), suggesting that LIP drives alpha/low-beta activity during periods of relatively worse visual-target detection.

We previously demonstrated coupling between theta phase and alpha-low/beta power in both mdPul and LIP (Supplementary

Figure 3). Because the reliability of phase estimates improves at higher power, changes in alpha/low-beta power as a function of theta phase (i.e., PAC) could create spurious relationships between spike–LFP phase coupling (in the alpha/low-beta range) and theta phase. We therefore conducted a control analysis, equating both the number of trials and alpha/low-beta power across theta-phase bins (see Methods). This stratification procedure did not change the results (Fig. 7b), confirming that the apparent directionality of connectivity between mdPul and LIP shifts with theta phase.

A similar pattern of results was revealed when we examined Granger causal influence as a function of theta phase (Fig. 6b). That is, the influence of mdPul on cortical hubs of the attention network was stronger during the "good" theta phase than during the "poor" theta phase. Supplementary Figure 7 shows the same results after stratification by alpha/low-beta power. These data again suggest that mdPul specifically regulates cortical activity during the theta-dependent attentional state associated with better visual-target detection (or enhanced perceptual sensitivity).

**The functional role of alpha/low-beta activity.** We previously demonstrated that the attentional state associated with better

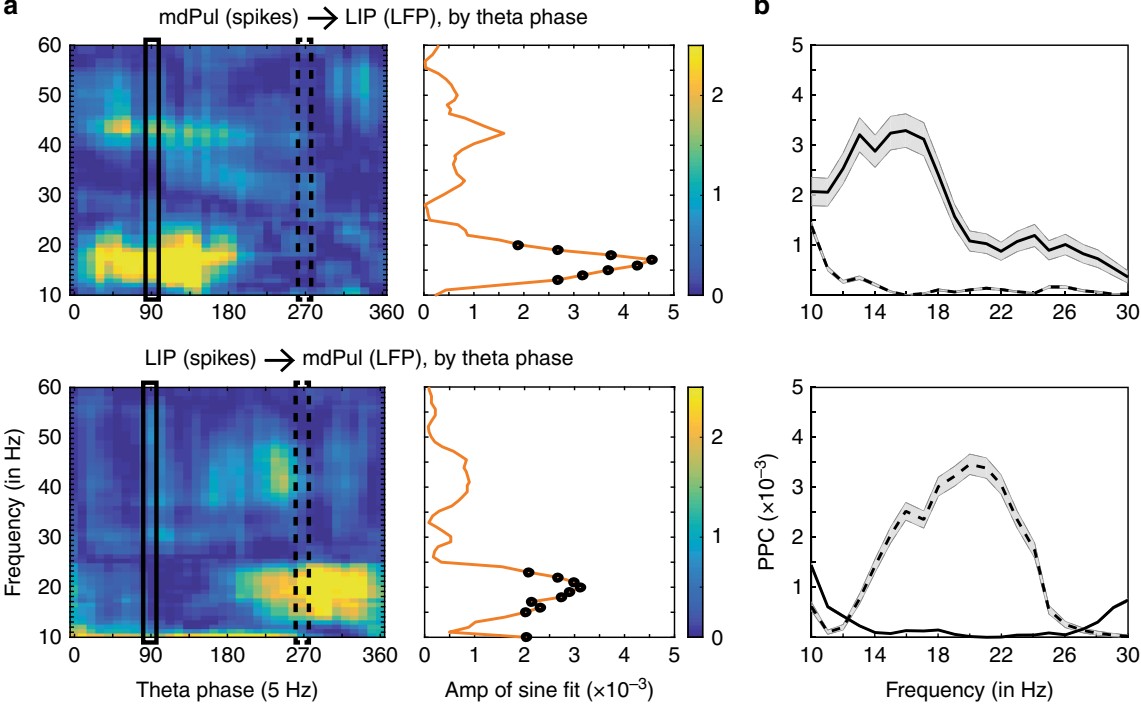

**Fig. 7** Spike–LFP phase coupling between mdPul and LIP as a function of theta phase. **a** Spike–LFP phase coupling (from 9 to 60 Hz) was calculated in overlapping theta-phase bins (on left), using step sizes of 10°. The resulting functions were then fit with one-cycle sine waves. The amplitude of these sine waves provided a measure of how strongly spike–LFP phase coupling was modulated by the phase of theta rhythms (on right, see Fig. 2A for depiction of a similar approach). The black dots represent statistically significant results after corrections for multiple comparisons. **b** To control for potentially spurious results from theta-dependent changes in alpha/low-beta power (i.e., PAC; see Supplementary Figure 3), a stratification procedure was used to equate trials numbers and alpha/low-beta power between two theta-phase bins, centered on either 90° or 270° (outlined in plots on far left). Because this stratification process involves downsampling trials and therefore different results on each run, it was re-run 1500 times. The above plots (**b**) represent the means and standard deviations (shaded regions around the lines) of those power-equating iterations

visual-target detection was characterized by an increase in LIP-dominated gamma activity[4]. A large body of evidence has linked such increases in cortical gamma to enhanced sensory processing[40]. Here, we investigated whether gamma power in LIP was dependent on the phase of alpha/low-beta activity during either of the theta-dependent attentional states. That is, we examined whether the primary frequency band for pulvino-cortical interactions (i.e., alpha/low-beta) influenced a frequency band previously associated with attention-related effects in LIP (i.e., gamma).

Figure 8a, b shows significant PAC between alpha/low-beta phase (at 15–18 Hz) and gamma power in LIP (28–49 Hz), occurring exclusively during periods of disengagement at the cued location (permutation test, $p < 0.001$). We previously reported that these periods of disengagement (i.e., the "poor" theta phase) were associated with lower overall gamma power in LIP (i.e., relative to periods of engagement at the cued location)[4]. Combined across the two studies, our findings suggest that alpha/low-beta activity disrupts gamma synchronization in LIP during periods of relatively worse visual-target detection (i.e., periods of disengagement), perhaps leading to lower overall gamma power during these periods (see ref. [4], Fig. 4).

We also measured significant PAC between alpha/low-beta phase and gamma power when receptive/response fields overlapped the non-cued location (Fig. 8a, b), occurring regardless of the theta-dependent attentional state (permutation test, $p < 0.001$). Alpha/low-beta activity in the cortex has been repeatedly linked to the suppression of sensory processing[41]. The present results (at both the cued and the non-cued location) are therefore consistent with a gating by inhibition hypothesis, whereby alpha/low-beta activity in LIP provides pulsed inhibition of attention-

related sensory processing[42]. This local disruption of gamma synchronization in LIP may also extend to between-region interactions. That is, state-specific increases in alpha/low-beta activity in LIP may similarly disrupt previously described gamma synchronization between LIP and FEF[4,41], further inhibiting attention-related sensory processing.

In comparison, we did not observe significant PAC between alpha/low-beta phase and gamma power in mdPul during either of the theta-dependent attentional states (Fig. 8c, d). These differences between mdPul and LIP in local PAC—as well as previously described links to opposite theta-dependent attentional states (Figs. 5 and 7 and Supplementary Figure 3)—suggest that alpha/low-beta activity has different functional roles in mdPul and LIP. Below we further discuss this apparent functional dissociation between mdPul-driven and LIP-driven alpha/low-beta activity (see the Discussion).

## Discussion

The present results are the first to functionally link mdPul with cortical hubs of the attention network. Locally, we demonstrated significantly increased spiking activity during the cue-target delay in mdPul (i.e., when receptive fields overlapped the cued location), which was similar in magnitude to that observed in LIP (Fig. 3). These electrophysiological results thus support previous findings from lesion and inactivation studies[19–23], indicating that mdPul plays a role in mediating spatial attention at behaviorally relevant locations. We further demonstrated increased coupling between mdPul and both FEF and LIP during spatial attention (Fig. 5), indicating that previously described anatomical connectivity between these cortical and subcortical structures[28–30]

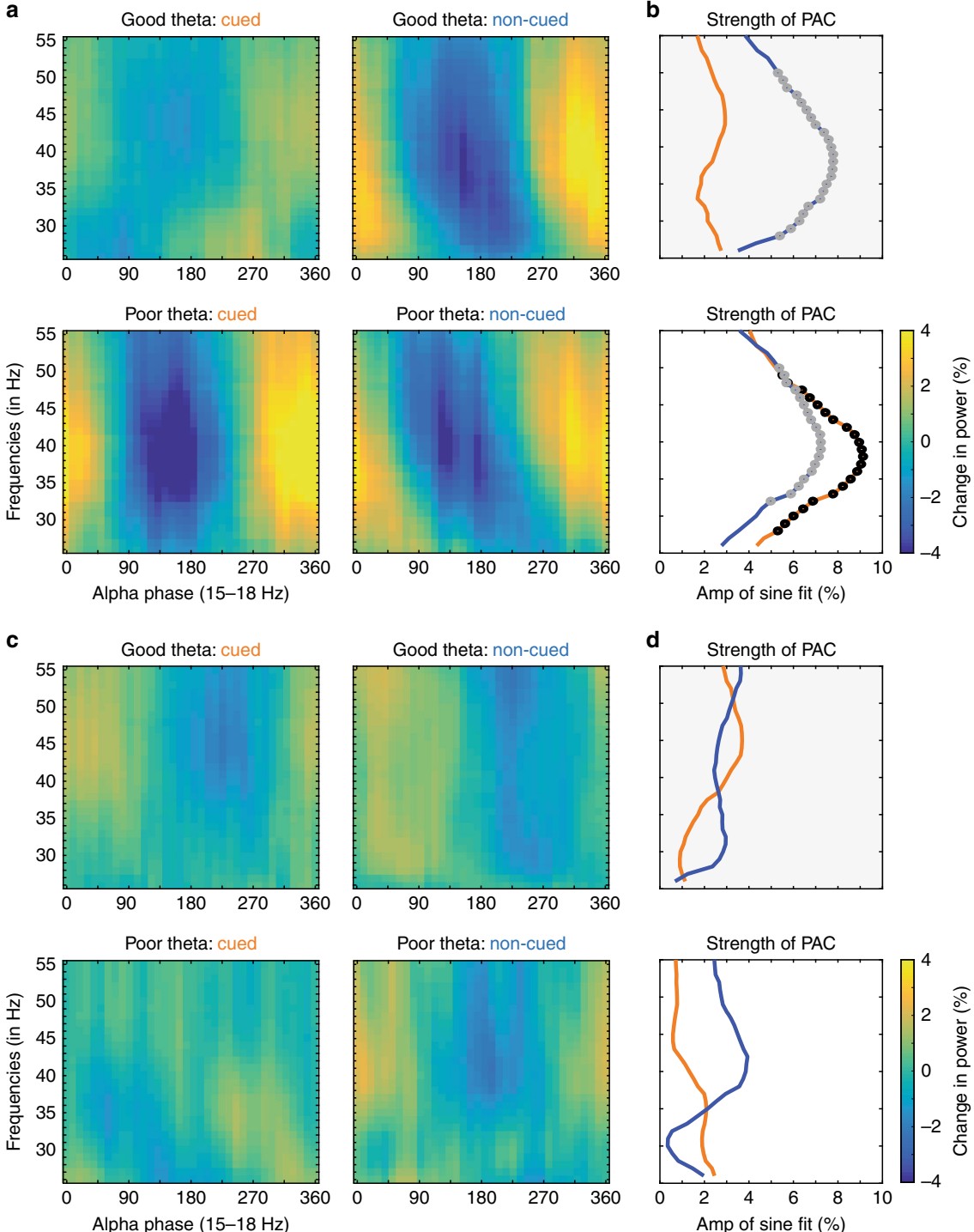

**Fig. 8** PAC in LIP between alpha/low-beta phase and gamma power is state-dependent. We measured phase–amplitude coupling after first binning trials based on the theta phase, with one bin centered on the "good" theta phase (i.e., the phase associated with better visual-target detection) and the other centered on the "bad" theta phase (i.e., the phase associated with worse visual-target detection). Panels **a**, **b** show power as a function of alpha/low-beta phase by theta-phase bin (good vs. poor) and by attention condition (cued [orange] vs. non-cued [blue]), for both **a** LIP ($N = 96$) and **b** mdPul ($N = 95$). **c**, **d** The strength of PAC was measured by using the fast Fourier transform to extract the amplitude of a one-cycle, sinusoidal component (see Fig. 2a for an illustration of this approach), for both **c** LIP and **d** mdPul. The black (cued location) and gray (uncued location) dots represent statistically significant results after corrections for multiple comparisons

serves to mediate attention-related processing. The present results thus firmly establish mdPul as a subcortical hub of the attention network. We next consider its functional role in this large-scale network.

Spatial attention samples the visual environment in theta-rhythmic cycles[3–8]. We previously demonstrated that these theta-rhythmic cycles are seemingly shaped by the phase of theta oscillations in frontal and parietal cortices[4,5]. Here, we demonstrate that theta activity in mdPul is similarly associated with alternating periods of either enhanced or diminished perceptual sensitivity (Fig. 4). That is, theta oscillations organize neural activity in both cortical and subcortical hubs of the attention

network into two rhythmically alternating attentional states. We propose that these attentional states alternately promote either (i) engagement at the presently attended location (and therefore enhanced perceptual sensitivity) or (ii) relative disengagement (and therefore diminished perceptual sensitivity), in anticipation of a potential attentional shift. Periods of engagement are associated with increased alpha/low-beta activity in mdPul, while periods of relative disengagement are associated with increased alpha/low-beta activity in LIP (Fig. 4; Supplementary Figure 3).

This state-dependent shifting of alpha/low-beta activity between mdPul and LIP is indicative of a functional dissociation. Although it is broadly assumed that there is functional specialization across hubs of the attention network[43,44], electrophysiological evidence has been sparse. Only a few studies have simultaneously recorded from multiple hubs of the attention network. Buschman and Miller[45], for example, reported that neural activity in FEF first signaled the target location during a serial search task (i.e., a task emphasizing goal-directed attention), while neural activity in LIP first signaled the target location during a pop-out task (i.e., a task emphasizing stimulus-driven attention). Their results thus indicated that FEF and LIP have task-specific functions, with (i) FEF leading LIP during goal-directed attention and (ii) LIP leading FEF during stimulus-driven attention. The present results instead provide evidence of functional specialization during unchanging task demands (i.e., during a task that promotes sustained attention at a cued location). We not only observed a rhythmic re-weighting of alpha/low-beta power between mdPul and LIP (Supplementary Figure 3), but spike–LFP phase coupling and Granger causality also revealed a rhythmic re-weighting of functional connectivity (Figs. 6 and 7). That is, the directionality of alpha/low-beta activity shifted between theta-dependent attentional states, with (i) mdPul regulating cortical activity during periods of engagement at the attended location (i.e., during the "good" theta phase) and (ii) LIP regulating thalamic activity during periods of relative disengagement (i.e., during the "poor" theta phase). These proposed functional roles for mdPul (i.e., engagement) and LIP (i.e., disengagement) mirror those first suggested by Posner and Petersen[43], largely based on data from human lesion studies.

Interactions between higher-order cortical regions and mdPul occurred almost exclusively in the alpha/low-beta band, which is typically linked to functional inhibition[41]. Both parietal cortex and the thalamus have been proposed as primary alpha/low-beta generators[15,46,47]. A recent study in human epilepsy patients investigated the source of alpha rhythms (7–13 Hz) during quiet wakefulness, recording from both posterior cortex and the pulvinar[46]. Halgren et al.[46] reported that alpha in higher-order cortical regions (i) propagates to visual-sensory cortices and (ii) leads alpha in the pulvinar. Their results thus support a cortical source of alpha rhythms. In comparison, Saalmann et al.[15] reported that alpha (8–15 Hz) in the ventral pulvinar leads alpha in visual-sensory cortices, specifically under conditions of spatial attention. Their results thus support a thalamic source of alpha rhythms. The present findings, on the other hand, indicate that the source of alpha/low-beta activity dynamically shifts between mdPul (i.e., thalamus) and LIP (i.e., cortex). That is, the source of alpha/low-beta rhythms is state-dependent (Figs. 6 and 7), potentially explaining the conflicting results of the previously described studies[15,46]. We next consider whether the function of alpha/low-beta rhythms in the attention network is the same regardless of the source (i.e., mdPul vs. LIP) and the theta-dependent attentional state (i.e., engagement vs. disengagement).

Electrophysiological studies in both humans and monkeys have repeatedly demonstrated that increased alpha/low-beta activity is a signature of suppressed processing in the sensory cortex[41,42,48]. That is, these studies show increased alpha/low-beta power in cortical regions processing task-irrelevant information[41]. For example, Worden et al.[49], who recorded electroencephalographic (EEG) data from humans during a spatial-cueing task, reported increased alpha power (8–14 Hz) over occipital cortex contralateral to the to-be-ignored hemifield (i.e., over cortex processing a task-irrelevant location). In agreement with this interpretation, we have proposed that increased alpha/low-beta power in LIP during periods of disengagement (i.e., during the "poor" theta phase) is associated with the suppression of sensory processing[4]. That is, we proposed that periodic increases in alpha power disrupt neural activity associated with processing at the presently attended location (Fig. 8a, b). This rhythmic re-weighting creates windows of opportunity when it is easier for an attentional shift to occur, if warranted by behavioral goals and the visual environment (e.g., stimulus salience). While previous research has generally linked theta rhythms to overt exploration, saccades in primates[50–52], and whisking in rodents[53,54], future studies will need to establish whether there is specifically an increased likelihood of covert or overt attentional shifts during theta-dependent periods of disengagement (i.e., the "poor" theta phase).

The suppressive role of alpha/low-beta activity in sensory cortex is well established, but only a few studies have measured alpha/low-beta activity in the pulvinar[15,16,46]. During periods of engagement at the attended location (i.e., during the "good" theta phase), we observed both (i) increased alpha/low-beta power in mdPul and (ii) increased pulvino-cortical influence, also occurring in alpha/low-beta (Figs. 6 and 7). Saalmann et al.[15] provided evidence that pulvino-cortical coordination in alpha/low-beta facilitates communication between regions of visual cortex (i.e., V4 and TEO) during spatial attention. Those results similarly demonstrated (i) stronger pulvino-cortical spike–LFP phase coupling and (ii) stronger pulvino-cortical Granger causal influence. Our results therefore indicate that mdPul plays a similar role in the attention network, coordinating alpha/low-beta activity in FEF and LIP (Fig. 5; Supplementary Figure 4). Notably, this coordination specifically occurs during the attentional state associated with engagement at the attended location (i.e., during the "good" theta phase), when alpha/low-beta power is relatively low in LIP (Supplementary Figure 3).

It remains unclear whether the increase in alpha/low-beta power in mdPul during periods of engagement is associated with functional inhibition (i.e., the functional role typically attributed to alpha/low-beta in the cortex). For example, this increase in alpha/low-beta power might be associated with a gating of indirect pathways (i.e., cortico-pulvino-cortical), emphasizing direct pathways (i.e., cortico-cortical) during periods of enhanced sensory processing (i.e., during the "good" theta phase). This hypothesis, however, conflicts with observed increases in pulvino-cortical influence during periods of engagement (Figs. 6 and 7). We therefore propose that alpha/low-beta activity in mdPul is associated with a different function than alpha/low-beta activity in the cortex. As further support for this proposal, Bollimunta et al.[55] previously described two cortical alpha generators, with opposite relationships to behavioral performance. For early visual cortices (i.e., V2 and V4), the primary alpha pacemaker was localized in the infragranular layer, and higher alpha power was associated with worse behavioral performance. For inferior temporal cortex, the primary alpha pacemaker was instead localized in the supragranular layer, and higher alpha power was associated with better behavioral performance. These results thus provide evidence that alpha/low-beta activity can reflect different functions in different brain regions, as here proposed for LIP and mdPul.

It should also be noted that cortex and the pulvinar have considerably different anatomical organizations. These

differences in anatomical organization may lead to differences in the strength and composition of LFPs, which arise from summed extracellular currents. Whereas cortex has a laminar and columnar organization, the pulvinar does not, instead being characterized by intermixed cell types with heterogeneous inputs, heterogeneous physiological properties, and relatively few local connections[18]. The neural mechanisms and cell types that generate alpha/low-beta activity in LIP and mdPul might therefore also be different.

In addition to further investigating the cell types and neural mechanisms underlying alpha/low-beta activity in LIP and mdPul, future studies will need to investigate the specific mechanisms through which increased alpha/low-beta activity in mdPul is associated with both (i) decreased alpha/low-beta activity in LIP and (ii) increased alpha/low-beta synchronization in FEF and LIP. That is, future studies should examine whether increased alpha/low-beta activity in mdPul plays a direct role in reducing alpha/low-beta power in LIP (and vice versa). Zhou et al.[16], for example, demonstrated that inactivating the ventral pulvinar increased the power of low-frequency oscillations in visual cortical regions, suggesting that normal pulvinar function (i.e., without inactivation) reduces low-frequency power. The pulvinar might therefore facilitate cortico-cortical communication by both synchronizing and reducing cortical alpha/low-beta activity.

The present findings demonstrate that theta-rhythmic sampling during spatial attention is characterized by changes in pulvino-cortical interactions. Theta-dependent shifts in alpha/low-beta activity reflect a functional re-weighting of the attention network, with (i) mdPul dominating alpha/low-beta activity during periods of engagement (and therefore enhanced perceptual sensitivity) and (ii) LIP dominating alpha/low-beta activity during periods of relative disengagement (and therefore diminished perceptual sensitivity). We propose that these state-dependent shifts in alpha/low-beta activity alternately favor brain regions and pathways associated with either (i) sampling at the presently attended location or (ii) shifting to another location. Theta rhythms thus seem to coordinate different functions of the attention network by shaping its spatiotemporal structure. Low-frequency oscillations might play a similar role in other large-scale networks, temporally resolving functional conflicts that arise, for example, from competing sensory stimuli[56] or from multiple items being held in working memory[57].

## Methods

**Subjects**. The study used two male *Macaca fascicularis* monkeys (6–9 years old). The Princeton University Animal Care and Use Committee approved all procedures, which conformed to the National Institutes of Health guidelines for the humane care and use of laboratory animals.

See Supplementary Figure 14 in ref. [4] and Supplementary Figure 8 in the present manuscript for demonstrations of common behavioral and electrophysiological effects between the two animals. Both monkeys demonstrated significantly better visual-target detection at the cued location, significantly increased spiking activity during the cue-target delay, significant theta-band rhythmicity in their behavioral data, and statistically significant PAC. Specific to the present manuscript, both monkeys also demonstrated theta-dependent patterns of functional connectivity (Supplementary Figure 14). We combined data from the two animals for all analyses presented in the main text.

**Behavioral task**. The monkeys performed a version of the Egly-Driver task (Fig. 2)[35], which was controlled using Presentation software (Fig. 2) (Ottawa Canada). We aborted trials if an animal's eye position deviated by more than one degree. An auditory "go" tone informed the animals when they could hold down a lever to begin a trial. We presented all visual stimuli on a 21-inch CRT monitor with a refresh rate of 100 Hz. Throughout each trial, the animals maintained fixation, focusing on a square at the center of the screen (eye-monitor distance = 57 cm). We monitored eye position using an infrared eye tracker (either an Eye-trac 6 at 240 Hz (Applied Science Laboratories Inc., Bedford MA) or an EyeLink 1000 Plus at 1000 Hz (SR Research Ltd), from central fixation). At the beginning of each trial there was a variable delay of 500–1200 ms. We next presented two-bar-

shaped objects (22° × 4.4°), oriented either horizontally or vertically, either above and below central fixation or to the left and right of central fixation. After a second variable delay of 500–1200 ms, we presented a brief spatial cue (100 ms) at the end of one of the bar-shaped objects. This cue indicated the most likely location for a subsequent, low-contrast (2.5–4%) visual target (with 78% cue validity). After a third variable delay (i.e., the cue-target delay) of 300–1600 ms, we presented a brief visual target (100 ms) at the end of one of the bar-shaped objects. The closest side of each bar-shaped object was 6.6° from central fixation and the closest corner of the target (4.4° × 4.4°) was 9.4° from central fixation. If the target did not occur at the cued location, it could occur at one of two non-cued locations (12% of all trials), either at the non-cued location on the same object as the cued location (i.e., the same-object location) or at the non-cued location on the second object (i.e., the different-object location). All of our analyses are focused on either the cued location (i.e., the attended condition) or the different-object location (i.e., our baseline condition). The monkeys released the lever in response to targets, regardless of the target location, receiving a juice reward for responses that occurred 150 to 650 ms after the target (i.e., for correct responses). For 10% of trials, we did not present a visual target. During these "catch trials," the monkeys released the lever when the screen cleared, which occurred 1600 ms after the cue.

**Electrophysiology**. We performed all surgical procedures under general anesthesia with isoflurane (induction 2–5%, maintenance 0.5–2.5%) and under strictly aseptic conditions. We affixed customized plastic recording chambers to the animals using both titanium skull screws and bone cement, with small craniotomies inside those chambers (4.5 mm diameter) providing access to our regions of interest (ROIs). Each craniotomy was fitted with a conical plastic-guide tube filled with bone wax[58]. These wax-filled guide tubes held glass-coated platinum–iridium electrodes (impedance: 5 MΩ) in place between recording sessions. For each animal, we first recorded from the left hemisphere and then moved the chambers to the right hemisphere for further recordings. Each recording session spanned a few hours, with up to seven sessions per week.

During each recording session, we used four thin rods that slid into hollows in the side of the implant to stabilize the animal's head. We independently lowered each electrode with microdrives (NAN Instruments Ltd, Nazaret Illit ISR) coupled to an adapter system that allowed different approach angles for each ROI. Electrode signals (40,000 Hz sample rate for spikes; 1000 Hz sample rate for LFPs) were amplified and filtered (150–8000 Hz for spikes; 0.7–300 Hz for LFPs) using a Plexon preamplifier (Plexon Inc., Dallas TX) with a high input impedance headstage and Multichannel Acquisition Processor (MAP) controlled by RASPUTIN software. We sorted spikes online to isolate neurons, and then re-sorted spikes for offline analyses using Plexon Offline Sorter software. The Egly-Driver task has four target locations, one in each quadrant. To determine where individual neurons and LFPs had their strongest responses, we compared responses to the spatial cue in each quadrant and during a quadrant-mapping task (i.e., large Gabor stimuli flashed in each quadrant).

Here, we present data from 95 recording sessions in the mediodorsal pulvinar (mdPul; monkey L, $N = 55$, monkey R, $N = 40$). We also simultaneously recorded from FEF and LIP. There were 51 recording sessions when mdPul and FEF had there strongest responses to stimuli in the same visual quadrant, 58 recording sessions when mdPul and LIP had their strongest responses in the same visual quadrant, and 67 recording sessions when FEF and LIP had their strongest responses to stimuli within the same visual quadrant (i.e., aligned RFs and/or multi-unit RFs). We used these recording sessions for between-region analyses. There were 31 recording sessions when all three ROIs had their strongest responses to stimuli within the same visual quadrant. We used these recording sessions to confirm that our between-region analyses of Granger causality were still valid when accounting for the influence of the third region (i.e., conditional Granger causality). Across all recording sessions, we isolated 52 task-responsive neurons in mdPul, 98 in FEF, and 98 in LIP. These neurons had significantly increased spike rates in response to the cue, the target, or both the cue and the target.

**Acquisition of MR images for electrode positioning**. We sedated the animals with ketamine (1–10 mg/kg i.m.) and xylazine (1–2 mg/kg i.m.), and also provided atropine (0.04 mg/kg i.m.). Sedation was maintained with tiletamine/zolazepam (1–5 mg/kg i.m.). We then placed the animals in an MR-compatible stereotaxic frame (1530 M; David Kopf Instruments, Tujunga CA) and monitored vital signs with wireless ECG and respiration sensors (Siemens AG, Berlin DEU) and a fiber optic temperature probe (FOTS100; Biopac Systems Inc, Goleta CA). We maintained body temperature with blankets and a warm water re-circulating pump (TP600; Stryker Corp., Kalamazoo, MI).

We collected structural MRI data for the whole brain on a Siemens 3 T MAGNETOM Skyra using a Siemens 11-cm loop coil placed above the head. A high-quality structural image was created for each animal by averaging 6–8 three-dimensional (3D) T1-weighted (T1w) volumes acquired in a single scan session (3D Magnetization-Prepared Rapid-Acquisition Gradient Echo (MPRAGE) sequence, voxel size: 0.5 mm, slice orientation: sagittal, slice thickness: 0.5 mm, field of view (FoV): 128 × 128 mm, FoV phase: 100%, repetition time (TR): 2700 ms, echo time (TE): 2.78 ms, inversion time (TI): 861 ms, base resolution: 256 × 256, acquisition time (TA): 11 min 31 s). T2-weighted (T2w) volumes were acquired with a 3D turbo spin echo with variable flip-angle echo trains (3D T2-SPACE)

sequence (voxel size: 0.5 mm, slice orientation: sagittal, slice thickness: 0.5 mm, FoV: 128 × 128 mm, FoV phase: 79.7%, TR: 3390 ms, TE: 386 ms, base resolution: 256 × 256, TA: 17 min 51 s. We used these T2w volumes both to select coordinates for chamber placements and to position electrodes for recordings. Platinum–iridium electrodes create a clearly identifiable, susceptibility-induced signal void along the length of the electrodes in structural MRI images. This "shadow" has a width of approximately one voxel (0.5 mm³ on either side of the electrode), allowing us to visualize electrode placement (Fig. 1).

Prior to recordings, we positioned electrodes just dorsal to our ROIs. The electrodes were then held in situ by customized guide tubes and lowered into the cortex over the course of typically 1 week of recordings. We then acquired additional structural MRI data prior to replacing the electrodes. We used the before and after images, as well as daily microdrive measurements, to reconstruct electrode tracks.

To further localize electrode penetrations, we aligned the D99 digital template atlas to each individual animal's high-quality T1w MRI volume, using a combination of FSL and AFNI software tools[59–61]. The D99 atlas is based on and aligned to MRI and histological data from the Saleem and Logothetis[62] atlas, and allows identification of labeled areas within the native 3D MRI volume of an individual animal. Briefly, we first extracted the brains from the T1w MRI volumes using the FSL brain extraction tool (BET)[63]. We then implemented the pipeline provided by Reveley et al.[61] to align the atlas to each monkey's MRI volume. This pipeline included a sequence of affine and nonlinear registration steps to first align the individual animal's MRI volume to the atlas, then inverting the transformations to warp the atlas to the animal's original native space. Once aligned, we overlaid the warped atlas' anatomical subdivisions upon the individual monkey's high-quality T1w MRI volume, co-registered the T2w MRI volumes that contained electrode penetrations, and visualized penetration locations with respect to the warped atlas on the animal's anatomy. For all recordings presented here, the electrodes were positioned in atlas-defined mdPul (labeled as medial pulvinar in the Saleem and Logothetis[62] atlas), FEF, and LIP. Figure 1 shows all of the mdPul penetrations. Fiebelkorn et al.[4] include representative penetrations into FEF and LIP.

**Spike rate.** For all analyses, we used a combination of customized MATLAB (MATLAB R2016a, The Mathworks Inc., Natick MA) functions and the Fieldtrip toolbox (http://www.ru.nl/neuroimaging/fieldtrip)[64]. We measured spike rates by convolving trial-level data with a Gaussian filter ($\sigma = 10$ ms) and then averaging across trials. We then used a non-parametric, randomization procedure to determine whether each neuron demonstrated an increased spike rate in response to either the cue or the target (i.e., within 250 ms after cue or target presentation). First, we randomly sampled a response-value from the pre-cue period ($-350$ to 0 ms) of each trial and averaged those response-values across trials to get a baseline spike rate. We then repeated this procedure 5000 times to generate a reference distribution. To generate a p-value, we determined the proportion of values in the reference distribution that exceeded the observed value from collected data. For all statistical comparisons, unless otherwise specified, we adopted an alpha criterion of 0.05, and used the Holm's sequential Bonferroni correction to control for multiple comparisons.

For population peri-stimulus time histograms (PSTHs), we grand-averaged spike rates across task-responsive neurons, after first normalizing the spike rate of each neuron based on its maximum response. For between-condition (i.e., cued vs. uncued) statistical comparisons, we first averaged the response across a pre-target window, spanning 500 ms prior to target presentation. We then conducted a Wilcoxon rank-sum test (Fig. 1).

**LFP power estimates.** To estimate LFP power in mdPul (Supplementary Figure 2), we used a fast Fourier transform (FFT) after first applying a Hanning window. Here, we examined power both before and after the cue was presented, restricting our analysis window to the 500 ms prior to either the cue or the target, and only including trials when the bar-cue and cue-target delay was at least 750 ms (i.e., to avoid the cue-evoked sensory response). In order to isolate oscillatory peaks from the 1/f background activity (i.e., to disentangle oscillatory from fractal components), we utilized irregular resampling (IRASA)[65]. IRASA takes advantage of the fact that irregular resampling of the neuronal signal by pairwise non-integer values slightly shifts the peak frequency of oscillatory signals by compressing or stretching the underlying signal. The 1/f component, in comparison, remains constant during resampling.

We used the results of IRASA to subtract 1/f estimates (i.e., the fractal component) from the original FFT results, isolating oscillatory residuals (Supplementary Figure 2). We then averaged our results across recording sessions. We restricted our analyses to frequencies below 60 Hz, as frequencies above this cut-off have been shown to reflect a broadband, non-oscillatory neural signal[66,67].

**Phase-detection relationships.** We investigated whether behavioral performance (i.e., hit rate) was associated with the oscillatory phase in the frequency-specific LFP signal, adapting an approach previously applied to EEG data[68]. We focused on the pre-target signal, convolving complex Morlet wavelets (from 3 to 60 Hz) with the LFP signal just prior to target presentation. We then derived pre-target phase estimates by taking the angle of the complex output.

We next sorted trials by phase and calculated HRs within overlapping, 180° phase bins (e.g., 0–180°). That is, we shifted our phase window in 5° steps (e.g., 5–185°, then 10–190°, etc.) until we generated phase-dependent HRs, spanning all phases for each frequency (Fig. 4a). This analysis created a function that represented the frequency-specific relationship between oscillatory phase and visual-target detection. We hypothesized a signature shape for this relationship, with a peak in visual-target detection separated from a trough by approximately 180°. Based on this hypothesis, we captured the strength of these phase-detection functions at each frequency by applying FFTs and keeping the second component. The absolute value of this second component represents the amplitude of an oscillation with a single cycle, matching the hypothesized shape (i.e., with a peak in visual-target detection separated from a trough by approximately 180°).

For statistical testing, we broke the relationship between phase and behavioral performance by shuffling the observed pre-target phase measurements relative to the observed behavioral outcomes (i.e., hits and misses). We re-shuffled the data 1500 times and repeated the analysis steps (as described above). We then compared the resulting reference distributions at each frequency to the observed phase-detection relationships (accounting for multiple comparisons).

The present results revealed a significant relationship between oscillatory phase in the theta band and the likelihood of visual-target detection. We previously reported such a dependency between theta phase in higher-order cortex (i.e., FEF and LIP) and phase-detection relationships at higher frequencies[4]. We therefore next examined whether phase-detection relationships in mdPul follow the same pattern, with phase-detection relationships at higher frequencies similarly being dependent on the phase of theta-band activity (at 5 Hz). We first binned trials into two theta-phase (180°) bins, the first centered on the peak in the previously measured phase-detection function (i.e., the "good" theta phase) and the second centered on the trough (i.e., the "poor" theta phase) in the previously measured phase-detection function (Fig. 4a). We then re-calculated the phase-detection relationships from 9 to 60 Hz, separately for each theta-phase bin (Fig. 4b).

To determine statistical significance, we used the same approach described above to break the relationship between phase and behavioral performance, but here we did it separately for each of the two theta-phase bins. This statistical analysis therefore determined whether there were significant links between visual-target detection and oscillatory phase at higher frequencies (from 9 to 60 Hz) within each theta-phase bin.

**Spike–LFP phase coupling.** We measured whether spikes in one brain region clustered at specific phases of the frequency-specific LFP signal (from 3 to 60 Hz) in the other region (e.g., spike times in FEF relative to oscillatory phase in LIP). As a measure of functional connectivity, spike–LFP phase coupling avoids spurious coupling associated with using a common recording reference, which can sometimes make it difficult to interpret spike–LFP phase coupling. For each spike time within a window from $-500$ to $-125$ ms prior to target presentation, we calculated corresponding phase estimates from the LFPs (centered on the spike time). We used the pairwise phase consistency (PPC) to measure spike–LFP phase coupling independent of differences in spike counts or trial numbers[69]. Higher PPC values indicate a greater clustering of spikes at specific phases of frequency-specific oscillatory activity in the LFP signal. For between-region analyses, increased spike–LFP phase coupling (as measured with PPC) is thought to reflect increased network connectivity. Although not a causal measure, spike–LFP phase coupling is also often interpreted with directionality, with spikes being interpreted as an output signal and LFPs being interpreted as an input signal[38].

For statistical testing, we (i) shuffled (1500 times) trials between conditions (cued vs. uncued) and (ii) re-calculated the difference in PPC between the shuffled conditions. We then compared the resulting reference distributions with the observed difference in PPC values.

We also measured spike–LFP phase coupling (from 9 to 60 Hz) as a function of theta phase (at 5 Hz). Here, we iteratively calculated spike–LFP phase coupling in overlapping, 180° phase windows (e.g., 0–180°), shifting the phase window in 5° steps (e.g., 5–185°, then 10–190°, etc.). To measure the strength of any link between theta phase and spike–LFP phase coupling at higher frequencies, we hypothesized a signature shape, with a peak in theta-dependent spike–LFP phase coupling separated from a trough by approximately 180° (i.e., the same signature shape that we previously hypothesized for relationships between oscillatory phase and the likelihood of visual-target detection). That is, we hypothesized that there would be a theta phase with particularly strong spike–LFP phase coupling (at higher frequencies), and 180° away from that phase, a theta phase with particularly weak spike–LFP phase coupling (at higher frequencies). Based on this hypothesis, we reduced theta-dependent, spike–LFP phase coupling functions to a single value for each frequency by applying FFTs and keeping the second component[68]. This second component represents the amplitude of an oscillation with a single cycle, matching our hypothesized shape (see Fig. 4a).

Higher oscillatory power is associated with more reliable phase estimates. Changes in power as a function of theta phase (i.e., PAC) could thus create spurious relationships between spike–LFP phase coupling and theta phase. We controlled for this relationship between power and phase estimates by conducting a control analysis that equated higher-frequency power between theta-phase bins. We compared theta-phase bins centered at 90° and 270°, using the ft_stratify function from the FieldTrip toolbox (Donders Institute for Brain, Cognition, and Behavior), which also equates sample sizes. A key element of this stratification

procedure is a subsampling of the original dataset to equate power, with this subsampling being somewhat different on each run. We therefore re-ran the stratification procedure 1500 times. Figure 7b displays the mean and standard deviation of these power-equating iterations, confirming the theta-dependent results shown in Fig. 7a.

**Granger causality**. We estimated the influence of the different ROIs on each other (e.g., the influence of mdPul on LIP) by measuring Granger causality when response fields in each ROI overlapped the cued location. We first downsampled the data to 250 Hz, subtracted the mean, and then divided by the standard deviation. For MVAR modeling, with the BSMART toolbox for MATLAB[70], we used a model order of 8, which generally corresponded to the first Akaike information criterion value. For paired estimates (e.g., mdPul vs. LIP), we averaged across all recording sessions when each of two ROIs had overlapping response fields. We also calculated conditional Granger causality for a subset of recording sessions when all three ROIs had overlapping response fields ($N = 31$). Conditional Granger causality measures the influence of one brain area (Y) on another (X), after taking into account additional areas (Z). The general pattern of results for Granger causal influence between ROIs (Fig. 6) remained the same when instead applying conditional Granger causality (Supplementary Figure 5).

We also examined Granger causal influence as a function of theta phase. We first used a two-cycle wavelet to measure the phase of theta-band activity just prior to target presentation. The wavelet was centered at −250 ms. We then sorted trials into two theta-phase bins (see Fig. 4a), one centered on the theta phase associated with enhanced perceptual sensitivity (i.e., the "good" theta phase) and the other centered on the theta phase associated with diminished perceptual sensitivity (i.e., the poor theta phase). For each theta-phase bin, we measured Granger causal influence using an epoch also centered at −250 ms (prior to target presentation), but only overlapping a time period equivalent to half of a theta cycle.

For statistical testing, we re-shuffled the trial data and re-calculated Granger causality for each of 1500 iterations. We then compared the compiled reference distribution of differences between, for example, mdPul to LIP and LIP to mdPul Granger causal influence to the observed values (controlling for multiple comparisons).

**Cross-frequency PAC**. We first examined PAC between FEF and LIP, focusing exclusively on the relationship between oscillatory phase in one region and high-frequency band (HFB) activity (from 80 to 200 Hz) in the other. HFB is an established proxy for population spiking[39]. We convolved complex Morlet wavelets with the LFP signal prior to target presentation (from −750 to −200 ms, in 10-ms steps), using the results to (i) derive phase estimates from 9 to 35 Hz (in 1-Hz steps) and (ii) to extract HFB activity. To get HFB, we first calculated power in 10-Hz steps (i.e., 80, 90, 100…). We then baseline corrected these power estimates by means of a z-score, relative to the pre-cue baseline, before averaging across HFB frequencies (from 80 to 200 Hz). Note that this approach accounts for the $1/f$ signal drop off in HFB at increasing frequencies.

We next sorted frequency-specific phase estimates (from 9 to 35 Hz) and calculated average HFB power within overlapping 180° phase bins (e.g., 0–180°), shifting the smoothing window in 5° steps (e.g., 5–185°, then 10–190°, etc.). To aid in visual comparisons across frequencies, we normalized these phase–power relationships by subtracting and dividing by the average power across all phases (separately for each frequency). We then multiplied by 100 to reveal the percent modulation in HFB power as a function of the oscillatory phase. The preceding steps (i.e., binning power estimates by phase) are similar to other approaches that measure PAC[71,72]. Here, we specifically hypothesized that PAC should have a signature shape, with a peak in PAC separated from a trough by approximately 180° (i.e., the same signature shape that we previously hypothesized for relationships between oscillatory phase and the likelihood of visual-target detection). We therefore reduced phase–power functions to a single value for each phase-providing frequency by applying the FFT to each function (i.e., at each frequency, from 3 to 60 Hz) and keeping the second component. This second component represents a one-cycle sine wave, matching the hypothesized shape of our phase-power functions. The amplitude of this one-cycle, sinusoidal component —determined both by how closely the function approximated a one-cycle sine wave and by the effect size—was used to measure the strength of PAC[4,68]. We used the same procedure to measure within-region PAC for mdPul (Supplementary Figure 3), specifically examining the link between theta phase (at 5 Hz) and higher-frequency power (from 9 to 60 Hz). We previously reported within-region PAC (from 9 to 60 Hz) for FEF and LIP[11]. We also used this procedure to examine PAC between alpha/low-beta phase and gamma power, separately for the "good" and "poor" theta-phase bins (Fig. 8).

For statistical testing, we (i) shuffled our observed pre-target phase estimates (1500 times) relative to observed power (breaking the relationship between phase and power) and (ii) repeated our analysis steps. We then compared the magnitude of resulting reference distributions to the observed PAC (accounting for multiple comparisons).

**Code availability**. All custom code used for this manuscript is available upon request from the Lead Contact: Ian C. Fiebelkorn.

## Data availability

All data contained in this manuscript are available upon request from the Lead Contact: Ian C. Fiebelkorn.

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

## Acknowledgements

This work was supported by a training fellowship to I.C.F. (F32EY023465), and by grants from NIMH (R01MH064063, Silvio O. Conte Center (1P50MH109429–01A1), NEI (RO1EY017699, R21EY023565), and the James S. McDonnell Foundation to S.K.

## Author contributions

Conceptualization: I.C.F. and S.K.; methodology: I.C.F., S.K., and M.A.P.; investigation: I. C.F. and M.A.P.; formal analysis: I.C.F.; resources: S.K.; funding acquisition: S.K.; writing —original draft: I.C.F. and S.K.; writing—review & editing: I.C.F., S.K., and M.A.P.

## Additional information

**Competing interests:** The authors declare no competing interests.

