## [Transparent Peer Review File · Nature Communications]

Reviewers' comments:

Reviewer #1 (Remarks to the Author):

In this manuscript, Fiebelkorn and colleagues examined interactions between the medial dorsal Pulvinar (mdPul), the FEF, and LIP in monkeys engaged in a spatial attention task. The analyses focused on the relationship between theta-rhythmic behavioral sampling, as previously demonstrated by the authors, and pulvino-cortical interactions. The results suggest that these interactions occur predominantly in the frequency range of 10-20 Hz, and functional connectivity shifts across the behavioral theta phase, with the mdPul as the source during attentional engagement and LIP as the source during disengagement. The findings provide the first demonstration of a functional link between the pulvinar and higher-order cortical association areas that have been proposed as essential hubs of the attention network. Further, these findings suggest a mechanism for functional specialization across this network. The manuscript is well-written, the statistical analyses are sound, and the findings should be of interest to a wide audience. I have only a couple of suggestions to improve clarity in a final version.

1. From this work, and the authors' previous findings, it is clear that the phase of low-frequency activity in mdPul (and FEF and LIP) is linked to the likelihood of target detection. However, it would be helpful to know the extent to which this result depends on the power of low-frequency oscillatory activity. Was there any evidence for increased power around 5 Hz in the LFP or was the behavior-phase relationship independent of any changes in low-frequency power?

2. In their previous work, (Fiebelkorn et al., 2018) the authors observed theta-dependent phase-detection relationships in LIP at 10-20 Hz (associated with poor theta) and 30-40 Hz (associated with good theta). However, the present manuscript focuses only on alpha/low-beta relationships. It would be helpful to clarify the choice of bands in the current paper and provide some explanation of how to interpret these findings relative to the previous findings by these authors. In particular, Figure S3 shows some evidence of PAC in LIP around 30 Hz for the good theta phase, at the similar phase to the 15-30 Hz PAC in mdPUL. In addition, PAC is seen most clearly at 15-30 Hz in mdPul (in the good theta phase) and at 10-15 Hz in LIP (in the bad theta phase). Accordingly, it would be helpful for the authors to discuss whether these functional specializations across the network are best understood as a change in source of the alpha/low-beta across distinct theta phases or rather as a shift in frequency.

Reviewer #2 (Remarks to the Author):

This study builds on earlier work from the Fiebelkorn, Kastner and colleagues, in which they showed evidence for rhythmic sampling during sustained attention in humans (Fiebelkorn et al., 2013) and non-human primates (Fiebelkorn et al., 2018). In the latter study, they concluded that this sampling is mediated by oscillations in the frontoparietal attentional network. Work from the Kastner laboratory (Saalmann et al., 2012) had also implicated the pulvinar in regulating synchronization between V4 and TEO. Here, they extend this line of research by recording from the MD nucleus of the pulvinar and two hubs of the fronto-parietal attentional control network (FEF and LIP) as monkeys performed an attention task in which attention was deployed to detect the appearance of a low contrast visual stimulus. Comparing trials in which attention was cued either into the receptive fields / response fields of neurons in each area vs. away to a second location, they find a variety of different patterns of interaction among these areas that relate systematically to behavior, including:

1. The monkey's performance on detecting the target varies (+/- 3%) with the phase of 5 Hz ("theta") oscillations in the LFP of the MD pulvinar.

2. In the advantaged theta phase, there is increased phase dependence in the 12-16 Hz range, but not in the disadvantaged theta phase.

3. Differential spike / lfp phase for mdPul-FEF (where spikes in mdPul were found to be phase locked to LFPs in FEF but not vice versa) and LIP, where the phase coupling was bidirectional.

Strengths:

The experiments are a tour de force, involving recording simultaneously not just from two brain areas (which is becoming more common) but three. This is the first study to record simultaneously from three different "hubs" of the attentional system, and helps establish a link between MD pulvinar and two cortical areas that have been implicated in numerous studies as being involved in attentional selection. The authors use appropriate nonparametric statistical tests (permutation tests) to assess the statistical significance of the various patterns they report. The study reveals a rich and complex description of various patterns, including differences in the phase coupling of neural signals (action potentials band pass filtered LFPs) between these areas.

Shortcomings:

One shortcoming of the paper is that it is pretty descriptive, without a clear testable motivating hypothesis. It documents a rich zoo of different phase coupling relationships across different pairs of the three areas studied, which are characterized across multiple frequency bands and directions of coupling, using a variety of different measures and statistics (spike-phase coupling, high frequency gamma power, granger causality, with trials binned based on theta phase, others).

The authors do provide, in the Discussion, an interpretation of the meaning of the various patterns they observe. They propose that the different phases of theta filtered LFP reflect attentional states that alternately promote either (i) engagement at the presently attended location (and therefore enhanced perceptual sensitivity) or (ii) relative disengagement (and therefore diminished perceptual sensitivity), in anticipation of a potential attentional shift. They further propose that MD Pulvinar plays a role in engaging attention while LIP plays a role in disengagement. But these are ad hoc interpretations, not a motivating falsifiable hypothesis that drives the work.

From this reviewer's perspective (which is admittedly biased by my interest in understanding the brain circuits underlying cognitive processes), the paper does not, yield clear insights into the mechanisms underlying attention. This is because the paper treats the different band pass filtered lfp patterns as though they were neural mechanisms. The LFP (filtered any way you like) is only indirectly related to the circuits that make up the brain, which are composed of networks of neurons, with their various complements of channels, receptors, neurotransmitters, membrane constants, patterns of connectivity, and so forth. These circuits interact with one another in ways that give rise to return currents and these combine to yield the extracellular potential that, when low pass filtered is the local field potential.

The LFP is thus an indirect measure of neural activity, but it is not (with the possible exception of ephaptic communication) a casual mechanism. It is certainly of interest (and may ultimately be illuminating) that different frequency bands vary systematically in relation to one another. This tells us that the dynamics of the underlying circuits vary with attentional state differently across FEF, LIP and the MD nucleus of the thalamus. But additional work would be needed to understand the implications of these observations for the underlying circuits. One would like to see a specific model proposed and tested. This would seem to require a model of the spiking activity of the circuits involved in the three areas and the circuits connecting them, coupled with a model linking this spiking activity to the return currents that give rise to the LFP, all coupled with a linking hypothesis connecting neural activity and perception. Why, for example, is the monkey's performance better in one phase of the theta filtered LFP than another?

Specific points:

The data are collapsed across monkeys. It would be good to see the data separately for each monkey to gauge how reliable this effect is, across animals.

The behavioral effects are fairly modest — a 6% variation in hit rate with phase at 5 Hz.

Stylistic:

The very first thing the reader needs to know is the task, which is not presented in the manuscript - the reader is directed to read another paper from the same group. The first data presented is also not in the manuscript — the reader is directed to the supplemental information. I appreciate the space limitations but found this disconcerting. Supplemental data should not be the first data the reader will want to see. The authors may want to consider introducing the supplemental data, which is not central to the paper, later.

At some points, there are abbreviations that are not defined, and the data are not clearly described. For example:

The authors introduce PAC without defining it ("Phase Advantaged Coupling", I guess?)

"We therefore measured whether theta phase in mdPul influenced the likelihood of detecting a low-contrast visual target (see online Methods)."

Phase of what? LFP, I assume, but the reader will be left wondering.

The text is at times a bit verbose:

Example 1: Referring to prior work from the Kastner lab the authors write: "There has been previous evidence that that (sic) the ventral pulvinar...".

"Has been"? Is the evidence no longer here? Simpler: "It has previously been shown that..." or "We have previously found that.."

Example 2: "specifically occurring under conditions of spatial attention " What does this mean? The entire experiment was performed while the animal engaged in an attention-demanding task. I think they simply mean "when attention was directed to the stimuli within the recorded neurons' receptive fields."

Reviewer #3 (Remarks to the Author):

The manuscript by Fiebelkorn et al. reports on the first physiological characterization of macaque (fascicularis) neuronal populations within the medial pulvinar (mdPulv) during simultaneous measurements of activity within two hubs of the cortical attention network, frontal eye fields (FEF) and lateral intraparietal area (LIP) during a spatial attention task. Recent work from this team provides evidence that rhythmic activity across FEF and LIP in the theta frequency (3-6 Hz) range plays a key role determining sensitivity of spatial attention. The present study seeks to examine the possible role of mdPulv in driving or participating in these theta rhythms. The results are interesting and important; the experiments identify similar theta gating of spatial attention within the mdPulv and importantly find evidence of phasic causal influence of mdPulv on LIP during periods of active attentional engagement. Overall the presentation of single unit and local field potential (LFP) data convincingly establish the main results: visually responsive neurons in mdPulv like those in FEF and LIP demonstrate delay period spike rate elevations associated with attentional engagement, a clear phase relationship of mdPulv theta rhythm cycle and target

detection is present similar to findings present in recent earlier studies of this phenomenon in FEF and LIP by the same investigative group, this theta cycle relationship centered around ~5Hz shows cycle dependent correlation with behavior similar to prior findings in cortex. Separating neuronal activity in “good” versus “poor” phases of theta cycle additional evidence for frequency multiplexing at higher frequencies within the alpha and low beta ranges is also identified using phase amplitude coupling measures which isolate the effect to the ‘good’ phase of the theta rhythm within mdPulv, an opposite findings to earlier measurements in LIP.

The study further examines the potential causal role for mdPulv in regulating the flow of activity within the three recorded structures. Examining spike-LFP correlations a bi-directional relationship of influences is established between mdPulv and LIP but not with FEF with respect to low frequency oscillatory activity; high frequency activity only showed relationships between LIP and mdPulv in the spike-LFP analyses. Using Granger Causality (GC) measures, the directionality of influence of activity across mdPulv, FEF and LIP is assessed with causal influence appearing more strongly for mdPulv as a modulator of alpha and low beta activity within cortical regions; additional measurements utilized very high frequency wide band (80-200Hz) activity as a proxy for spiking activity. Comparing spike-LFP coupling during the different ‘good’ and ‘poor’ phases of theta, spikes and alpha/low beta activity showed that spikes in mdPulv coupled to alpha/low beta LFP of LIP and FEF during the good phase suggesting a driving effect during this period of enhanced target detection. An opposite finding of relationship for spikes driving mdPulv during ‘poor’ phases was demonstrated. Finally, additional analyses examine the relationship of gamma activity within LIP and uncover evidence of alternation of alpha/low beta activity disrupting gamma synchronization along the lines of alpha power increases thought to reflect local inhibition in EEG experiments of human attention cited in the discussion.

Overall, the manuscript provides a detailed and extremely interesting study of the relationships of different components of neuronal activity within mdPulv in comparison to simultaneous activity within FEF and LIP. These findings should be of significant interest to neuroscientists studying cerebral networks supporting attentional function. The findings draw further attention to the organizing role of theta rhythms across different brain networks and raise many questions about the other potential sources or interactions that may exist with other brain structures.

Minor points:

1. The LFP within mdPulv should have different biophysical properties than the columnar structures of LIP and FEF given the radial organization of thalamic dendritic arbors, have the investigators considered any potential influence of these local measurement differences with respect to the results?
2. The relationships between FEF and mdPulv lack the closeness in reciprocity of functional interactions seen in the mdPulv and LIP measurements. Have the investigators considered other candidates of frontal cortical structures that may influence mdPulv more causally during “good” theta phases? The pulvinar receives a wide range of inputs and it would seem unlikely that the behavioral state control for the task rests locally within this subcortical structure?
3. The idea that the ‘poor’ phase of theta cycle is only for flexible disengagement seems somewhat less compelling than the evidence for the ‘good’ reflecting standard optimal attentional function as consistent with other studies at many levels. Along these lines is possible that there is something like an alteration of local spatial and the monkey equivalent of brief mind-wandering or internal focusing as is proposed for anti-correlation in time of human resting state networks measured in fMRI? The shift to LIP influence on mdPulv spike-LFP during ‘poor’ phases seems potentially consistent with such a possibility.
4. The title seems a little to indefinite for the forces of the data interpretation. Something like “Medial pulvinar theta rhythm modulates/switches attentional states in macaque”...might be more catchy.

Reviewer #1 (Remarks to the Author):

In this manuscript, Fiebelkorn and colleagues examined interactions between the medial dorsal Pulvinar (mdPul), the FEF, and LIP in monkeys engaged in a spatial attention task. The analyses focused on the relationship between theta-rhythmic behavioral sampling, as previously demonstrated by the authors, and pulvino-cortical interactions. The results suggest that these interactions occur predominantly in the frequency range of 10-20 Hz, and functional connectivity shifts across the behavioral theta phase, with the mdPul as the source during attentional engagement and LIP as the source during disengagement. The findings provide the first demonstration of a functional link between the pulvinar and higher-order cortical association areas that have been proposed as essential hubs of the attention network. Further, these findings suggest a mechanism for functional specialization across this network. The manuscript is well-written, the statistical

analyses are sound, and the findings should be of interest to a wide audience. I have only a couple of suggestions to improve clarity in a final version.

We thank the reviewer for his/her encouragement and helpful suggestions.

1. From this work, and the authors' previous findings, it is clear that the phase of low-frequency activity in mdPul (and FEF and LIP) is linked to the likelihood of target detection. However, it would be helpful to know the extent to which this result depends on the power of low-frequency oscillatory activity. Was there any evidence for increased power around 5 Hz in the LFP or was the behavior-phase relationship independent of any changes in low-frequency power?

*Here, the reviewer asks two questions. First, was there increased power in the LFP around 5 Hz? In response, we have included a new supplemental figure (Fig. S2), which shows power in mdPul from 3–60 Hz, both based on the 500 ms prior to cue presentation and based on the 500 ms prior to target presentation. We used irregular sampling (IRASA, Wen & Liu, Brain Topography, 2016) to remove the 1/f component from the FFTs. There was a general decrease in low-frequency power following the spatial cue, including power in the theta range. There was, however, a peak in the theta range that exceeded the 1/f component of the LFP signal, during both the baseline period (i.e., pre-cue) and the cue-target delay (i.e., pre-target). The new supplemental figure is pasted below. We now refer to this supplemental figure in the results section (page 8): **“Figure S2 shows LFP power in mdPul, from 3–60 Hz, demonstrating a clear peak in the theta range (after removing the 1/f component). We next measured whether theta phase in the LFP influenced the likelihood of detecting a low-contrast visual target (see Methods).”***

Figure S2. Power spectral density in mdPul when response fields overlapped the cued location, prior to the cue (black line) and during the cue-target delay (orange line). **(A)** There was a general drop in low-frequency power following the cue, but **(B)** removing the fractal (or $1/f$) component, using the IRASA approach (Wen & Liu, 2016), demonstrates an apparent oscillatory peak in the theta range during both trial periods, as well as a peak in the alpha/low-beta range. **(C)** The oscillatory components for each recording session, prior to the cue (on left) and during the cue-target delay (on right), relative to the average across sessions (bolded lines).

Second, the reviewer asks if the phase-behavior relationship in mdPul, at approximately 5 Hz, is dependent on theta power? Here, we split trials into two bins based on the median power, for each frequency from 3–10 Hz. As was the case for the original analysis—which was based on all the trials—the phase-detection plots for both high- and low-power bins showed a peak in the theta range. This peak remained statistically significant in the low-power bin, despite using only half the trials. For the high-power bin, the p -value was 0.08. Phase-detection relationships in the theta range therefore appear to be robust to differences in theta power. A reviewer figure is pasted below:

Reviewer Figure 1. Phase-detection relationships in the theta range are robust to binning by power. That is, we split trials into two bins based on median power and recalculated phase-detection relationships in each bin, from 3–10 Hz. In each bin, we observed a peak in the theta range. This peak was statistically significant in the low-power bin ($p = 0.05$), but not in the high-power bin ($p = 0.08$).

2. In their previous work, (Fiebelkorn et al., 2018) the authors observed theta-dependent phase-detection relationships in LIP at 10-20 Hz (associated with poor theta) and 30-40 Hz (associated with good theta). However, the present manuscript focuses only on alpha/low-beta relationships. It would be helpful to clarify the choice of bands in the current paper and provide some explanation of how to interpret these findings relative to the previous findings by these authors.

Here, we primarily focused on alpha/low-beta activity because effects in mdPul and between mdPul and higher-order regions occurred almost exclusively in the alpha/low-beta range (see also Saalman et al., 2012, Science). As the reviewer points out, our previous work (Fiebelkorn et al., 2018, Neuron) highlights the importance of both alpha/low-beta and gamma activity in LIP. The present manuscript shows that these two frequency bands are coupled in LIP (through phase-amplitude coupling). Specifically, we show that alpha/low-beta phase modulates gamma power (Fig. 6) during relative disengagement at the cued location (i.e., during the “poor” theta phase). We previously reported that periods of engagement (i.e., the “good” theta phase) at the cued location were characterized by an increase in gamma activity in LIP. Alpha/low-beta activity during periods of relative disengagement seems to periodically disrupt this gamma activity, consistent with alpha/low-beta playing a role in sensory gating (see Jensen & Mazaheri, 2010, Frontier in Human Neuroscience). Increased alpha/low-beta activity in LIP may also disrupt the previously described gamma-synchronization between LIP and FEF (Fiebelkorn et al., 2018, Neuron). We have added text to the manuscript to clarify this connection with our previous work. The section of the manuscript that describes the

state-specific relationship between alpha/low-beta phase and gamma power is pasted below, with the new text in bold.

From the Results Section (page 14 of the manuscript):

“Figure 8A–B shows significant PAC between alpha/low-beta phase (at 15–18 Hz) and gamma power in LIP (28–49 Hz), occurring exclusively during periods of disengagement at the cued location (permutation test, $p < 0.001$). We previously reported that these periods of disengagement (i.e., the “poor” theta phase) were associated with lower overall gamma power in LIP (i.e., relative to periods of engagement at the cued location)⁴. Combined across the two studies, our findings suggest that alpha/low-beta activity disrupts gamma synchronization in LIP during periods of relatively worse visual-target detection (i.e., periods of disengagement), perhaps leading to lower overall gamma power during these periods (see Fiebelkorn, et al.⁴, Fig. 4).

*We also measured significant PAC between alpha/low-beta phase and gamma power when receptive/response fields overlapped the non-cued location (Fig. 6A–B), occurring regardless of the theta-dependent attentional state (permutation test, $p < 0.001$). Alpha/low-beta activity in cortex has been repeatedly linked to the suppression of sensory processing⁴². The present results (at both the cued and the non-cued location) are therefore consistent with a gating by inhibition hypothesis, whereby alpha/low-beta activity in LIP provides pulsed inhibition of attention-related sensory processing⁴³. **This local disruption of gamma synchronization in LIP may also extend to between-region interactions. That is, state-specific increases in alpha/low-beta activity in LIP may similarly disrupt previously described gamma synchronization between LIP and FEF^{4,41}, further inhibiting attention-related sensory processing.**”*

In particular, Figure S3 shows some evidence of PAC in LIP around 30 Hz for the good theta phase, at the similar phase to the 15-30 Hz PAC in mdPUL. In addition, PAC is seen most clearly at 15-30 Hz in mdPul (in the good theta phase) and at 10-15 Hz in LIP (in the bad theta phase). Accordingly, it would be helpful for the authors to discuss whether these functional specializations across the network are best understood as a change in source of the alpha/low-beta across distinct theta phases or rather as a shift in frequency.

In Figure S3, there is significant PAC between theta phase and alpha/low-beta power for both LIP (9–16 Hz) and mdPul (11–23 Hz), but the peaks in PAC occur at 12 Hz and 18 Hz, respectively. Figure 7 similarly suggests that alpha/low-beta activity in mdPul—occurring exclusively during the “good” theta phase—might have a higher peak frequency than alpha/low-beta activity in LIP.

While there seems to be a change in the brain region driving alpha/low-beta activity across the two attentional states (i.e., from mdPul during periods of engagement at the cued location to LIP during periods of disengagement), our data cannot speak to

whether the neural mechanisms or cell types generating alpha/low-beta activity differ depending on whether the source is LIP or mdPul. We agree that this is a critical question for future research. There are considerable differences between the anatomical organizations of LIP and mdPul, suggesting that there might be different neural mechanisms underlying alpha/low-beta activity in the two structures. Whereas cortex has a laminar and columnar organization, the pulvinar has no laminar or columnar organization, instead being characterized by intermixed cell types with heterogeneous physiological properties and relatively few local connections. Differences in the neural mechanisms underlying alpha/low-beta activity in LIP and mdPul may explain why this frequency band seems to be associated with different functional roles in the two brain regions. We hope that our future studies will contribute to unraveling this issue. For example, we are presently conducting recordings with multi-contact probes. With more single units in the pulvinar, we hope to link alpha/low-beta activity with specific functionally defined cell types. Our discussion of these issues in the present manuscript has been pasted below, with new text in bold

From the Discussion Section (page 19 of the manuscript):

“It remains unclear whether the increase in alpha/low-beta power in mdPul during periods of engagement is associated with functional inhibition (i.e., the functional role typically attributed to alpha/low-beta in cortex). For example, this increase in alpha/low-beta power might be associated with a gating of indirect pathways (i.e., cortico-pulvino-cortical), emphasizing direct pathways (i.e., cortico-cortical) during periods of enhanced sensory processing (i.e., during the “good” theta phase). This hypothesis, however, conflicts with observed increases in pulvino-cortical influence during periods of engagement (Figs. 4, 5). We therefore propose that alpha/low-beta activity in mdPul is associated with a different function than alpha/low-beta activity in cortex. As further support for this proposal, Bollimunta, et al.⁵⁰ previously described two cortical alpha generators, with opposite relationships to behavioral performance. For early visual cortices (i.e., V2 and V4), the primary alpha pacemaker was localized in the infragranular layer, and higher alpha power was associated with worse behavioral performance. For inferior temporal (IT) cortex, the primary alpha pacemaker was instead localized in the supragranular layer, and higher alpha power was associated with better behavioral performance. These results thus provide evidence that alpha/low-beta activity can reflect different functions in different brain regions, as here proposed for LIP and mdPul.

It should also be noted that cortex and the pulvinar have considerably different anatomical organizations. These differences in anatomical organization may lead to differences in the strength and composition of LFPs, which arise from summed extracellular currents. Whereas cortex has a laminar and columnar organization, the pulvinar does not, instead being characterized by intermixed cell types with heterogeneous inputs, heterogeneous physiological properties, and relatively few local connections (Bridge, Leopold, Bourne, 2016, TICS). The neural mechanisms and cell

types that generate alpha/low-beta activity in LIP and mdPul might therefore also be different.”

Reviewer #2 (Remarks to the Author):

This study builds on earlier work from the Fiebelkorn, Kastner and colleagues, in which they showed evidence for rhythmic sampling during sustained attention in humans (Fiebelkorn et al., 2013) and non-human primates (Fiebelkorn et al., 2018). In the latter study, they concluded that this sampling is mediated by oscillations in the frontoparietal attentional network. Work from the Kastner laboratory (Saalmann et al., 2012) had also implicated the pulvinar in regulating synchronization between V4 and TEO. Here, they extend this line of research by recording from the MD nucleus of the pulvinar and two hubs of the fronto-parietal attentional control network (FEF and LIP) as monkeys performed an attention task in which attention was deployed to detect the appearance of a low contrast visual stimulus. Comparing trials in which attention was cued either into the receptive fields / response fields of neurons in each area vs. away to a second location, they find a variety of different patterns of interaction among these areas that relate systematically to behavior, including:

1. The monkey's performance on detecting the target varies (\pm 3%) with the phase of 5 Hz ("theta") oscillations in the LFP of the MD pulvinar.
2. In the advantaged theta phase, there is increased phase dependence in the 12-16 Hz range, but not in the disadvantaged theta phase.
3. Differential spike / lfp phase for mdPul-FEF (where spikes in mdPul were found to be phase locked to LFPs in FEF but not vice versa) and LIP, where the phase coupling was bidirectional.

Strengths:

The experiments are a tour de force, involving recording simultaneously not just from two brain areas (which is becoming more common) but three. This is the first study to record simultaneously from three different "hubs" of the attentional system, and helps establish a link between MD pulvinar and two cortical areas that have been implicated in numerous studies as being involved in attentional selection. The authors use appropriate nonparametric statistical tests (permutation tests) to assess the statistical significance of the various patterns they report. The study reveals a rich and complex description of various patterns, including differences in the phase coupling of neural signals (action potentials band pass filtered LFPs) between these areas.

We thank the reviewer for his/her acknowledgement of the difficulty in conducting these multisite recordings and for the positive feedback.

Shortcomings:

One shortcoming of the paper is that it is pretty descriptive, without a clear testable motivating hypothesis. It documents a rich zoo of different phase coupling relationships across different pairs of the three areas studied, which are characterized across multiple frequency bands and directions of coupling, using a variety of different measures and statistics (spike-phase coupling, high frequency gamma power, granger causality, with trials binned based on theta phase, others).

Here, we respectfully disagree with the reviewer. We hypothesized that the mdPul coordinates cortical activity in the attention network (e.g., Saalman et al., 2012, Science). More specifically, we hypothesized that mdPul plays a role in coordinating the theta-rhythmic sampling (e.g., stated in both the abstract and in the introduction on page 4, top of third paragraph) that we and others have recently shown to be characteristic of spatial attention (e.g., Fiebelkorn et al., 2018, Neuron; Helfrich et al., 2018, Neuron). Based on these hypotheses, our analyses were designed to do the following: (i) determine whether mdPul is a functional hub of the attention network—these were the first recordings in mdPul during an attention-related task (stated on page 6, first paragraph), (ii) determine whether neural activity in mdPul, like that in FEF and LIP, is linked to behaviorally relevant theta-band activity (stated on page 8, first paragraph), (iii) determine whether network interactions between mdPul and higher-order cortical regions are shaped (or influenced by) theta-rhythmic sampling (stated on page 12, first paragraph), and (iv) determine whether the nature of these network interactions (e.g., the specific frequency band, apparent directionality) and their relationships to behavior can shed light on functional role of mdPul (this is the focus of the last analysis in the results section and a primary focus of the discussion, where we synthesize our findings).

The authors do provide, in the Discussion, an interpretation of the meaning of the various patterns they observe. They propose that the different phases of theta filtered LFP reflect attentional states that alternately promote either (i) engagement at the presently attended location (and therefore enhanced perceptual sensitivity) or (ii) relative disengagement (and therefore diminished perceptual sensitivity), in anticipation of a potential attentional shift. They further propose that MD Pulvinar plays a role in engaging attention while LIP plays a role in disengagement. But these are ad hoc interpretations, not a motivating falsifiable hypothesis that drives the work.

Again, we respectfully disagree with the reviewer's characterization of our work. This work was undertaken with the goal of better understanding the functional role of mdPul in the attention network, specifically during theta-rhythmic sampling. We think that the present work is a terrific starting point. We also acknowledge, of course, that there is much more work to be done, as is true for all insightful science. Our results are consistent with mdPul being a functional hub of the attention network, playing a critical

role in theta-rhythmic sampling during spatial attention. The mdPul organizes cortical activity exclusively during the theta phase associated with better visual-target detection. Posner & Petersen (1990, Annual Review of Neuroscience) first proposed specialized roles for mdPul in attentional engagement and LIP in attentional disengagement, based on patient studies. Our work is consistent with their classical hypothesis and begins to provide a neural basis for it.

From this reviewer's perspective (which is admittedly biased by my interest in understanding the brain circuits underlying cognitive processes), the paper does not, yield clear insights into the mechanisms underlying attention. This is because the paper treats the different band pass filtered lfp patterns as though they were neural mechanisms. The LFP (filtered any way you like) is only indirectly related to the circuits that make up the brain, which are composed of networks of neurons, with their various complements of channels, receptors, neurotransmitters, membrane constants, patterns of connectivity, and so forth. These circuits interact with one another in ways that give rise to return currents and these combine to yield the extracellular potential that, when low pass filtered is the local field potential. The LFP is thus an indirect measure of neural activity, but it is not (with the possible exception of ephaptic communication) a casual mechanism.

Whereas the BOLD signal, for example, can be characterized as an indirect measure of neural activity, the LFP is a direct measure of neural activity. We would agree, however, that it is an indirect measure of neural communication. That being said, we used the toolkit typically employed in systems neuroscience in order to measure network connectivity. In lieu of being able to detect and record from synaptically linked single neurons across a large-scale, anatomically separated network—which is unfeasible at this point—spike-LFP phase coupling provides a reasonable, often-used measure of whether two brain regions are functionally interconnected. The same between-region measures used in the present manuscript have been used in numerous published studies, leading to important and illuminating results. Here are a few such studies within the field of attention research: Buschman & Miller, 2007, Science; Gregoriou et al., 2009, Science; Saalmann et al., 2012, Science.

It is certainly of interest (and may ultimately be illuminating) that different frequency bands vary systematically in relation to one another. This tells us that the dynamics of the underlying circuits vary with attentional state differently across FEF, LIP and the MD nucleus of the thalamus.

Yes, establishing that the dynamics of the underlying circuits vary within rhythmically alternating, behaviorally relevant attentional states is a critical step toward understanding the neural basis of attention.

But additional work would be needed to understand the implications of these observations for the underlying circuits. One would like to see a specific model

proposed and tested. This would seem to require a model of the spiking activity of the circuits involved in the three areas and the circuits connecting them, coupled with a model linking this spiking activity to the return currents that give rise to the LFP, all coupled with a linking hypothesis connecting neural activity and perception.

We certainly agree that additional work will need to be done to better understand the underlying neural mechanisms. To that end, we are presently recording data with multi-contact probes, which will potentially allow us to resolve cortical layers and will give us a much greater number of within-session single units. We plan to examine, for example, how different cortical layers and cell types are associated with different oscillatory frequencies in the LFP signal, providing further clues regarding the functional significance of those oscillatory frequencies. We are also collaborating with Nancy Kopell at Boston University to develop models for how these rhythms are produced by the underlying circuits. However, all of that is outside the scope of the present study.

Why, for example, is the monkey's performance better in one phase of the theta filtered LFP than another?

We start with the observation and careful examination of behavioral performance. We ask 'what is a specific behavior'? And then we study neural signals in relation to that specific behavior. That is, we investigate which neural mechanisms and signals can best predict the behavioral outcome. The neural basis of a particular behavior may also illuminate why it exists. We have elaborated on the 'why' in a forthcoming TICS paper.

The data are collapsed across monkeys. It would be good to see the data separately for each monkey to gauge how reliable this effect is, across animals.

We previously demonstrated monkey-specific behavioral evidence of theta-rhythmic sampling during spatial attention (Fiebelkorn et al., 2018, Neuron, Fig. S14). In further response to the reviewer, the present manuscript now includes a supplemental figure (Fig. S8) that shows several key findings for each monkey: (i) conditions-specific delay spiking, (ii) coupling between theta phase and alpha/low-beta power, and (iii) Granger causal influence by theta phase (good vs. bad theta phase). The new figure is pasted below:

Figure S8. Both monkeys (L and R) contributed to effects that we observed after combining data across the two animals. For example, **(A)** shows greater delay spiking when receptive fields overlapped the cued location (in orange), relative to when receptive fields overlapped the non-cued location (in blue), **(B)** shows significant coupling between theta phase (at 5Hz) and alpha/low-beta power, and **(C)** shows theta-dependent changes in functional connectivity, with greater Granger causal influence from mdPul to higher-order cortex (i.e., FEF and LIP) during the “good” theta phase (i.e., during periods of enhanced perceptual sensitivity). The black dots in **(B)** represent statistical significance after corrections for multiple comparisons.

The behavioral effects are fairly modest — a 6% variation in hit rate with phase at 5 Hz.

We appreciate the reviewer’s viewpoint, but we have a different perspective. This behavioral effect is either comparable to or stronger than previously reported findings that linked oscillatory phase with behavioral performance (e.g., Mathewson et al., Journal of Neuroscience, 2009; Busch et al., Journal of Neuroscience, 2009; Busch & VanRullen, PNAS, 2010; Landau et al., Current Biology, 2015), and for several reasons, it likely underrepresents the true influence of theta phase on hit rates. For example, calculating hit rates in a sliding window (90 degree bins) smoothed the data and therefore decreased the effect size. Moreover, we would argue that a 6 percentage point shift in hit rate is a strong effect, considering, for example, that it is (1) linked to the

phase of a single oscillatory frequency that is itself multiplexed with higher frequencies, and that it is (2) based on neural activity from a single hub of the much larger network that contributes to attention-related sampling.

Stylistic:

The very first thing the reader needs to know is the task, which is not presented in the manuscript - the reader is directed to read another paper from the same group. The first data presented is also not in the manuscript — the reader is directed to the supplemental information. I appreciate the space limitations but found this disconcerting. Supplemental data should not be the first data the reader will want to see. The authors may want to consider introducing the supplemental data, which is not central to the paper, later.

The reviewer makes a good point, and we have room in the main text for additional figures, so we now include both (i) a representation of the recording penetrations, targeting the mediodorsal pulvinar (Fig. 1, was Fig. S1) and (ii) a schematic of the task (now Fig.2). The new figure, which depicts the task, is pasted below.

Figure 2. A schematic of the behavioral task. The animals pressed a lever to begin each trial and maintained central fixation. A spatial cue indicated, with 78% cue validity, where the visual-target was most likely to occur. Following a variable cue-target delay, a low-contrast visual target was either presented at the cued location or at a non-cued location. The animals responded by releasing the lever. We also included catch trials to track false alarms.

At some points, there are abbreviations that are not defined, and the data are not clearly described. For example:

The authors introduce PAC without defining it ("Phase Advantaged Coupling", I guess?)

We thank the reviewer for catching this. We mistakenly did not define PAC (phase-amplitude coupling) until the second usage. We have fixed this. It is also defined in the figure legends.

"We therefore measured whether theta phase in mdPul influenced the likelihood of detecting a low-contrast visual target (see online Methods)."

Phase of what? LFP, I assume, but the reader will be left wondering.

We have edited the text to make it clearer that we are measuring oscillatory phase in the LFP signal. The text now states (on page 8 of the manuscript): "we next measured whether theta phase in the LFP influenced the likelihood of detecting a low-contrast visual target (see Methods)."

The text is at times a bit verbose:

Example 1: Referring to prior work from the Kastner lab the authors write: "There has been previous evidence that that (sic) the ventral pulvinar...".

"Has been"? Is the evidence no longer here? Simpler: "It has previously been shown that..." or "We have previously found that.."

Example 2: "specifically occurring under conditions of spatial attention " What does this mean? The entire experiment was performed while the animal engaged in an attention-demanding task. I think they simply mean "when attention was directed to the stimuli within the recorded neurons' receptive fields."

We have carefully gone through the manuscript looking for places, like the above two examples, where we could be more precise in our language. We have edited the text where appropriate.

Reviewer #3 (Remarks to the Author):

The manuscript by Fiebelkorn et al. reports on the first physiological characterization of macaque (fascicularis) neuronal populations within the medial pulvinar (mdPulv) during simultaneous measurements of activity within two hubs of the cortical attention network, frontal eye fields (FEF) and lateral intraparietal area (LIP) during a spatial attention task. Recent work from this team provides evidence that rhythmic activity across FEF and LIP in the theta frequency (3-6 Hz) range plays a key role determining sensitivity of spatial attention. The present study seeks to examine the possible role of mdPulv in driving or participating in these theta rhythms. The results are interesting and important; the experiments identify similar theta gating of spatial attention within the

mdPulv and importantly find evidence of phasic causal influence of mdPulv on LIP during periods of active attentional engagement.

Overall the presentation of single unit and local field potential (LFP) data convincingly establish the main results: visually responsive neurons in mdPulv like those in FEF and LIP demonstrate delay period spike rate elevations associated with attentional engagement, a clear phase relationship of mdPulv theta rhythm cycle and target detection is present similar to findings present in recent earlier studies of this phenomenon in FEF and LIP by the same investigative group, this theta cycle relationship centered around ~5Hz shows cycle dependent correlation with behavior similar to prior findings in cortex. Separating neuronal activity in “good” versus “poor” phases of theta cycle additional evidence for frequency multiplexing at higher frequencies within the alpha and low beta ranges is also identified using phase amplitude coupling measures which isolate the effect to the ‘good’ phase of the theta rhythm within mdPulv, an opposite findings to earlier measurements in LIP.

The study further examines the potential causal role for mdPulv in regulating the flow of activity within the three recorded structures. Examining spike-LFP correlations a bi-directional relationship of influences is established between mdPulv and LIP but not with FEF with respect to low frequency oscillatory activity; high frequency activity only showed relationships between LIP and mdPulv in the spike-LFP analyses. Using Granger Causality (GC) measures, the directionality of influence of activity across mdPulv, FEF and LIP is assessed with causal influence appearing more strongly for mdPulv as a modulator of alpha and low beta activity within cortical regions; additional measurements utilized very high frequency wide band (80-200Hz) activity as a proxy for spiking activity. Comparing spike-LFP coupling during the different ‘good’ and ‘poor’ phases of theta, spikes and alpha/low beta activity showed that spikes in mdPulv coupled to alpha/low beta LFP of LIP and FEF during the good phase suggesting a driving effect during this period of enhanced target detection. An opposite finding of relationship for spikes driving mdPulv during ‘poor’ phases was demonstrated. Finally, additional analyses examine the relationship of gamma activity within LIP and uncover evidence of alternation of alpha/low beta activity disrupting gamma synchronization along the lines of alpha power increases thought to reflect local inhibition in EEG experiments of human attention cited in the discussion.

Overall, the manuscript provides a detailed and extremely interesting study of the relationships of different components of neuronal activity within mdPulv in comparison to simultaneous activity within FEF and LIP. These findings should be of significant interest to neuroscientists studying cerebral networks supporting attentional function. The findings draw further attention to the organizing role of theta rhythms across different brain networks and raise many questions about the other potential sources or interactions that may exist with other brain structures.

We thank the reviewer for his/her positive and detailed assessment of our work, as well as his/her constructive feedback.

Minor points:

1. The LFP within mdPulv should have different biophysical properties than the columnar structures of LIP and FEF given the radial organization of thalamic dendritic arbors, have the investigators considered any potential influence of these local measurement differences with respect to the results?

This is certainly a very important point: there are key differences between the anatomical organization of cortex and the pulvinar. As a result, the neural mechanisms and cell types underlying, for example, alpha-band activity might likewise be different. More generally, differences in anatomical organization may lead to differences in the composition of LFPs, which result from summed extracellular currents. This is related to a comment that was made by Reviewer #1. Our discussion of these issues in the present manuscript has been pasted below, with the new text in bold.

From the Discussion Section (page 19 of the manuscript):

“It remains unclear whether the increase in alpha/low-beta power in mdPul during periods of engagement is associated with functional inhibition (i.e., the functional role typically attributed to alpha/low-beta in cortex). For example, this increase in alpha/low-beta power might be associated with a gating of indirect pathways (i.e., cortico-pulvino-cortical), emphasizing direct pathways (i.e., cortico-cortical) during periods of enhanced sensory processing (i.e., during the “good” theta phase). This hypothesis, however, conflicts with observed increases in pulvino-cortical influence during periods of engagement (Figs. 4, 5). We therefore propose that alpha/low-beta activity in mdPul is associated with a different function than alpha/low-beta activity in cortex. As further support for this proposal, Bollimunta, et al.⁵⁰ previously described two cortical alpha generators, with opposite relationships to behavioral performance. For early visual cortices (i.e., V2 and V4), the primary alpha pacemaker was localized in the infragranular layer, and higher alpha power was associated with worse behavioral performance. For inferior temporal (IT) cortex, the primary alpha pacemaker was instead localized in the supragranular layer, and higher alpha power was associated with better behavioral performance. These results thus provide evidence that alpha/low-beta activity can reflect different functions in different brain regions, as here proposed for LIP and mdPul.

It should also be noted that cortex and the pulvinar have considerably different anatomical organizations. These differences in anatomical organization may lead to differences in the strength and composition of LFPs, which arise from summed extracellular currents. Whereas cortex has a laminar and columnar organization, the pulvinar does not, instead being characterized by intermixed cell types with heterogeneous inputs, heterogeneous physiological properties, and relatively few local connections (Bridge, Leopold, Bourne, 2016, TICS). The neural mechanisms and cell types that generate alpha/low-beta activity in LIP and mdPul might therefore also be different.”

2. The relationships between FEF and mdPulv lack the closeness in reciprocity of functional interactions seen in the mdPulv and LIP measurements. Have the investigators considered other candidates of frontal cortical structures that may influence mdPulv more causally during “good” theta phases? The pulvinar receives a wide range of inputs and it would seem unlikely that the behavioral state control for the task rests locally within this subcortical structure?

For the present investigation, we only found evidence of pulvinar-to-FEF influence and not vice versa. As the reviewer correctly points out, FEF is not the only region of frontal cortex that is interconnected with mdPul (e.g., see Romanski et al., The Journal of Comparative Neurology, 1997). It therefore remains possible that different regions of frontal cortex causally influence mdPul. Given the existing literature in monkeys (e.g., work from Tirin Moore that demonstrates a causal role of FEF in spatial attention) and the single-contact recordings employed for the present investigation, FEF was arguably the most sensible target in frontal cortex. However, we certainly agree that future investigations—for example, pairing multi-contact cortical arrays with pulvinar recordings—need to include additional PFC regions.

3. The idea that the ‘poor’ phase of theta cycle is only for flexible disengagement seems somewhat less compelling than the evidence for the ‘good’ reflecting standard optimal attentional function as consistent with other studies at many levels. Along these lines is possible that there is something like an alteration of local spatial and the monkey equivalent of brief mind-wandering or internal focusing as is proposed for anti-correlation in time of human resting state networks measured in fMRI? The shift to LIP influence on mdPulv spike-LFP during ‘poor’ phases seems potentially consistent with such a possibility.

Here, the reviewer provides an alternative interpretation for what is happening during the “poor” theta phase. That is, that this phase is perhaps associated with a loss of external focus and brief mind wandering. This is not inconsistent with our ideas. For example, disengagement during the “poor” phase might involve re-allocating attention at a different external source such as a location in visual space, or re-allocating attention from external to internal sources (see Chun et al., 2011, Annual Review of Psychology). Our proposal that theta rhythms in the attention network temporally isolate the sensory and motor processes associated with environmental sampling is described in much more detail in a review paper that is presently under consideration at TICS. However, we agree with the reviewer that, for now, there is stronger data to support our interpretation of what is happening during the “good” theta phase than to support our interpretation of what is happening during the “poor” theta phase. We have therefore added the following sentence to the Discussion section (page 18):

“While previous research has generally linked theta rhythms to overt exploration, saccades in primates⁵⁰⁻⁵² and whisking in rodents^{53,54}, future studies will need to

establish whether there is specifically an increased probability of covert or overt attentional shifts during theta-dependent periods of disengagement (i.e., the “poor” theta phase).”

4. The title seems a little to indefinite for the forces of the data interpretation. Something like “Medial pulvinar theta rhythm modulates/switches attentional states in macaque” ...might be more catchy.

We thank the author for suggesting that we change the title. We agree that the previous title was indefinite, and we feel that the following title better describes the manuscript:

“The mediodorsal pulvinar coordinates the macaque fronto-parietal network during rhythmic spatial attention”

REVIEWERS' COMMENTS:

Reviewer #1 (Remarks to the Author):

The authors have addressed all of my concerns in this revised manuscript. I have no further concerns.

Reviewer #2 (Remarks to the Author):

I find the authors responses to the points I raised persuasive. I have no further comments.

Reviewer #3 (Remarks to the Author):

The authors have nicely addressed all of the issues that I had previous raised in review.